# Circular RNA *HMGCS1* sponges *MIR4521* to aggravate type 2 diabetes-induced vascular endothelial dysfunction

**Ming Zhang**[1], **Guangyi Du**[1], **Lianghua Xie**[1], **Yang Xu**[1], **Wei Chen**[1,2]*

[1]Department of Food Science and Nutrition, College of Biosystems Engineering and Food Science, Zhejiang University, Hangzhou, China; [2]Ningbo Innovation Center, Zhejiang University, Ningbo, China

**\*For correspondence:**
zjuchenwei@zju.edu.cn

**Competing interest:** The authors declare that no competing interests exist.

**Abstract** Noncoding RNA plays a pivotal role as novel regulators of endothelial cell function. Type 2 diabetes, acknowledged as a primary contributor to cardiovascular diseases, plays a vital role in vascular endothelial cell dysfunction due to induced abnormalities of glucolipid metabolism and oxidative stress. In this study, aberrant expression levels of *circHMGCS1* and *MIR4521* were observed in diabetes-induced human umbilical vein endothelial cell dysfunction. Persistent inhibition of *MIR4521* accelerated development and exacerbated vascular endothelial dysfunction in diabetic mice. Mechanistically, *circHMGCS1* upregulated arginase 1 by sponging *MIR4521*, leading to decrease in vascular nitric oxide secretion and inhibition of endothelial nitric oxide synthase activity, and an increase in the expression of adhesion molecules and generation of cellular reactive oxygen species, reduced vasodilation and accelerated the impairment of vascular endothelial function. Collectively, these findings illuminate the physiological role and interacting mechanisms of *circHMGCS1* and *MIR4521* in diabetes-induced cardiovascular diseases, suggesting that modulating the expression of *circHMGCS1* and *MIR4521* could serve as a potential strategy to prevent diabetes-associated cardiovascular diseases. Furthermore, our findings provide a novel technical avenue for unraveling ncRNAs regulatory roles of ncRNAs in diabetes and its associated complications.

## eLife assessment

This study presents **important** findings linking circHMGCS1 and miR-4521 in diabetes-induced vascular endothelial dysfunction. Overall, the evidence supporting the claims of the authors is **convincing**. The work will be of interest to biomedical scientists working with cardiovascular and/or RNA biology, particularly those studying diabetes.

## Introduction

Cardiovascular disease (CVD) is the leading cause of morbidity and mortality in patients with type 2 diabetes mellitus (T2DM). The incidence of CVD in T2DM individuals was estimated to be at least two to four times higher than that in non-diabetic individuals (*Gregg et al., 2016*; *Yun and Ko, 2021*). T2DM gives rise to abnormal metabolic conditions, including chronic hyperglycemia, dyslipidemia, insulin resistance, inflammation, and oxidative stress (*Roden and Shulman, 2019*). These processes ultimately lead to vascular endothelial damage, which disrupts the maintenance of vascular homeostasis, thereby promoting the development of CVD (*Lundberg and Weitzberg, 2022*; *Niemann et al., 2017*). Vascular endothelial dysfunction (VED) is considered the underlying basis for vascular complications at all stages of T2DM (*Eelen et al., 2015*; *Meigs et al., 2004*). However, the mechanism

by which the key molecules trigger endothelial dysfunction in diabetes-induced vascular impairment remains unknown.

Endothelial cells form a continuous monolayer lining the arterial, venous, and lymphatic vessels, playing a crucial physiological role in maintaining vascular homeostasis (*Bazzoni and Dejana, 2004*). Moreover, endothelial cells actively regulate vascular tone, preserve endothelial integrity, and modulate platelet activity, contributing to hemostasis and endothelial repair (*Li et al., 2019*). The leading cause of T2DM is a prolonged high-sugar, high-fat diet (HFHG), commonly known as a Western diet, which inevitably leads to VED, characterized by reduced content of nitric oxide (NO), uncoupling of endothelial nitric oxide synthase (*ENOS*), increased formation of reactive oxygen species (ROS) and upregulation of endothelial-related adhesion molecules such as intercellular adhesion molecule 1 (*ICAM1*), vascular cell adhesion molecule 1 (*VCAM1*), and endothelin 1 (*ET-1*). These alterations ultimately lead to elevated blood pressure and impaired vascular arteriolar relaxation, contributing to CVD development (*Xu et al., 2021*; *Yeh et al., 2022*; *Zheng et al., 2018*). Despite this understanding, the identification of pivotal regulatory factors governing gene expression in diabetes-induced VED remains a formidable challenge.

circRNAs are generated through back-splicing of precursor messenger RNAs (pre-mRNAs), exhibiting features such as a covalently closed loop structure, high stability, conservation, and tissue-specific expression. These molecules primarily function as miRNA sponges, regulators of transcription, and interactors with proteins (*Kristensen et al., 2019*; *Liu and Chen, 2022*). Conversely, miRNAs are single-stranded small RNA molecules of 21–23 nucleotides, characterized by high conservation, temporal and tissue-specific expression, and relatively poor stability. miRNAs—critical regulators of gene expression—control a wide array of cellular processes (*Gebert and MacRae, 2019*). The interplay between miRNAs and circRNAs forms a complex regulatory network, profoundly influencing diverse biological pathways and disease states. miRNAs serve as guides, targeting specific mRNAs to induce their degradation or translational repression, thereby suppressing gene expression post-transcriptionally (*Treiber et al., 2019*). By contrast, circRNAs contain miRNA binding sites, competitively inhibiting miRNA activity and thereby reducing miRNA binding to other mRNAs. This competitive adsorption effect modulates intracellular miRNA levels, affecting miRNA-mediated regulation of other target genes (*Cheng et al., 2019b*; *Huang et al., 2019*; *Shen et al., 2019*; *Zeng et al., 2021*).

The expression levels of circRNA in diabetic patients exhibit abnormalities, and its potential as a biomarker for early diagnosis or therapeutic target has been gradually being substantiated (*Jiang et al., 2022*; *Stoll et al., 2020*; *Tian et al., 2018*). However, research on circRNA in the context of diabetes-induced VED is scarce. Current research has predominantly focused on the regulatory roles of circRNA in diabetic cardiomyopathy (*Yuan et al., 2023*), diabetic nephropathy (*Yang and Liu, 2022*), gestational diabetes (*Du et al., 2022*), and diabetic retinopathy (*Zhu et al., 2019*). Investigations into circRNA regulation of VED have only described the phenomenon and analyzed the correlation of the results, lacking the precision to infer causation accurately (*Jiang et al., 2020*; *Liu et al., 2017*; *Ma et al., 2023*). Moreover, comprehension of VED induced by the complex multifactorial processes of diabetes pathogenesis remains relatively inadequate. Additionally, the interaction between circRNA and miRNA in diabetes-induced VED remains a subject of research controversy (*Cheng et al., 2019a*; *Liu et al., 2019*; *Pan et al., 2018*). Therefore, a more comprehensive research approach is required to elucidate the specific mechanisms by which circRNA operates in diabetes-related CVDs.

The present study elucidated the mechanisms underlying the interaction between circRNA and miRNA in regulating diabetes-associated VED. Through the screening strategy, we successfully identified *circHMGCS1*—a circRNA originating from the pre-mRNA of *HMGCS1*. We demonstrated that the upregulation of *circHMGCS1* and downregulation of *MIR4521* significantly promoted diabetes-induced VED. Moreover, in-depth mechanistic investigations revealed that *circHMGCS1* functioned as a *MIR4521* sponge, increasing arginase 1 (*ARG1*) expression in vascular endothelial cells and accelerating VED progression. Furthermore, the expression patterns and interaction mechanisms between *circHMGCS1* and *MIR4521* in endothelial cells were identified for the first time in these findings, which also highlight the *circHMGCS1/MIR4521/ARG1* axis as a novel therapeutic target for interventions designed to protect patients from diabetes-induced VED.

## Results

### Global expression analysis of endotheliocyte circRNAs

To explore the potential role of circRNAs in regulating endotheliocyte function, we characterized the transcriptome of circRNA derived from total RNA in human umbilical vein endothelial cells (HUVECs) stimulated by high palmitate and high glucose (PAHG), which allowed us to investigate the potential role of circRNAs in diabetes-induced VED (*Figure 1—figure supplement 1A–C*). A total of 17,179 known circRNAs were identified, with 66 of them exhibiting differential expression (GEO submission: GSE237597). Our main focus was on circRNAs that shared identical genomic and splicing lengths that exhibited fold changes greater than 2 or less than –2 (p<0.01) in PAHG-treated HUVECs (*Figure 1A*). Specifically, 48 circRNAs were upregulated, whereas 18 were downregulated (*Figure 1—figure supplement 1D*). Furthermore, approximately 66.65% of circRNAs were found to be produced through reverse splicing of exons (*Figure 1—figure supplement 1E*).

The observed changes in circRNA levels were confirmed through the real-time quantitative reverse transcription PCR (qRT-PCR) analysis of the five most upregulated circRNAs, suggesting that the results of the RNA-seq data are credible. Among them, *circHMGCS1* (hsa_circ_0008621, 899) nt in length, identified as *circHMGCS1* in subsequent studies because of its host gene being *HMGCS1* (*Figure 1—figure supplement 1F*; *Liang et al., 2021*) exhibited significant upregulation compared with the non-PAHG-treated condition (*Figure 1B*). The *circHMGCS1* sequence is situated on chromosome 5 of the human genome, specifically at 43292575–43297268 with no homology to mouse sequences. Subsequently, divergent primers were designed to amplify *circHMGCS1*, whereas convergent primers were designed to amplify the corresponding linear mRNA from cDNA and genomic DNA (gDNA) in HUEVCs. amplification of *circHMGCS1* was observed in cDNA but not in gDNA (*Figure 1C*), which confirmed the circularity of *circHMGCS1*. Sanger sequencing of the amplified product of *circHMGCS1* confirmed its sequence alignment with the annotated *circHMGCS1* in circBase (*Figure 1D*). This observation confirms the origin of *circHMGCS1* from exons 2–7 of the *HMGCS1* gene (*Figure 1E* and *Figure 1—figure supplement 1G, H*).

RNA-Fluorescence in situ hybridization (RNA-FISH) confirmed the robust expression of cytoplasmic *circHMGCS1* in HUVECs (*Figure 1F*). The qRT-PCR analysis of nuclear and cytoplasmic RNA revealed the predominant cytoplasmic expression of *circHMGCS1* in HUVECs (*Figure 1G*). qRT-PCR also demonstrated that *circHMGCS1* displayed a stable half-life exceeding 24 hr, whereas the linear transcript *HMGCS1* mRNA had a half-life of less than 8 hr (*Figure 1H*). Furthermore, *circHMGCS1* exhibited resistance to digestion by ribonuclease R (an exonuclease that selectively degrades linear RNA *Xiao and Wilusz, 2019*, also known as RNase R), whereas linear *HMGCS1* mRNA was easily degraded upon RNase R treatment (*Figure 1I*). These findings indicate that the higher expression of *circHMGCS1* in VED is more than just a byproduct of splicing and suggestive of functionality.

### Upregulation of circHMGCS1 promotes diabetes-induced VED

To explore the potential role of *circHMGCS1* in regulating endothelial cell function, we cloned exons 2–7 of *HMGCS1* into lentiviral vectors for ectopic overexpression of *circHMGCS1* (*Figure 2—figure supplement 1*). We found that *circHMGCS1* was successfully overexpressed in HUVECs without significant changes in *HMGCS1* mRNA expression, and the level of the circular transcripts increased nearly 60-fold than the level of the linear transcripts (*Figure 2A*). These results confirm that the circularization is efficient and that *circHMGCS1* has no effect on *HMGCS1* expression. Further experiments demonstrated that the overexpression of *circHMGCS1* stimulated the expression of adhesion molecules (*VCAM1*, *ICAM1*, and *ET-1*; *Figure 2B, C*), suggesting that *circHMGCS1* is involved in VED promotion. We then used PAHG to stimulate HUVEC-overexpressing *circHMGCS1*. NO levels were significantly decreased after PAHG stimulation for 24 hr, and *circHMGCS1* overexpression further decreased NO levels (*Figure 2D*), indicating that increased *circHMGCS1* expression inhibits endothelial diastolic function. *ENOS* activity was inhibited by PAHG, and upregulated *circHMGCS1* expression further decreased *ENOS* activity (*Figure 2E*). Superoxide is one of the major ROS known to promote atherosclerosis (*Griendling et al., 2016*). Therefore, we used DHE to assess superoxide levels in endothelial cells. The overexpression of *circHMGCS1* enhanced PAHG-induced ROS generation (*Figure 2F*), increasing oxidative stress in endothelial cells. The expression levels of adhesion molecules (*VCAM1*, *ICAM1*, and *ET-1*) were further elevated with *circHMGCS1* overexpression (*Figure 2G, H*). These

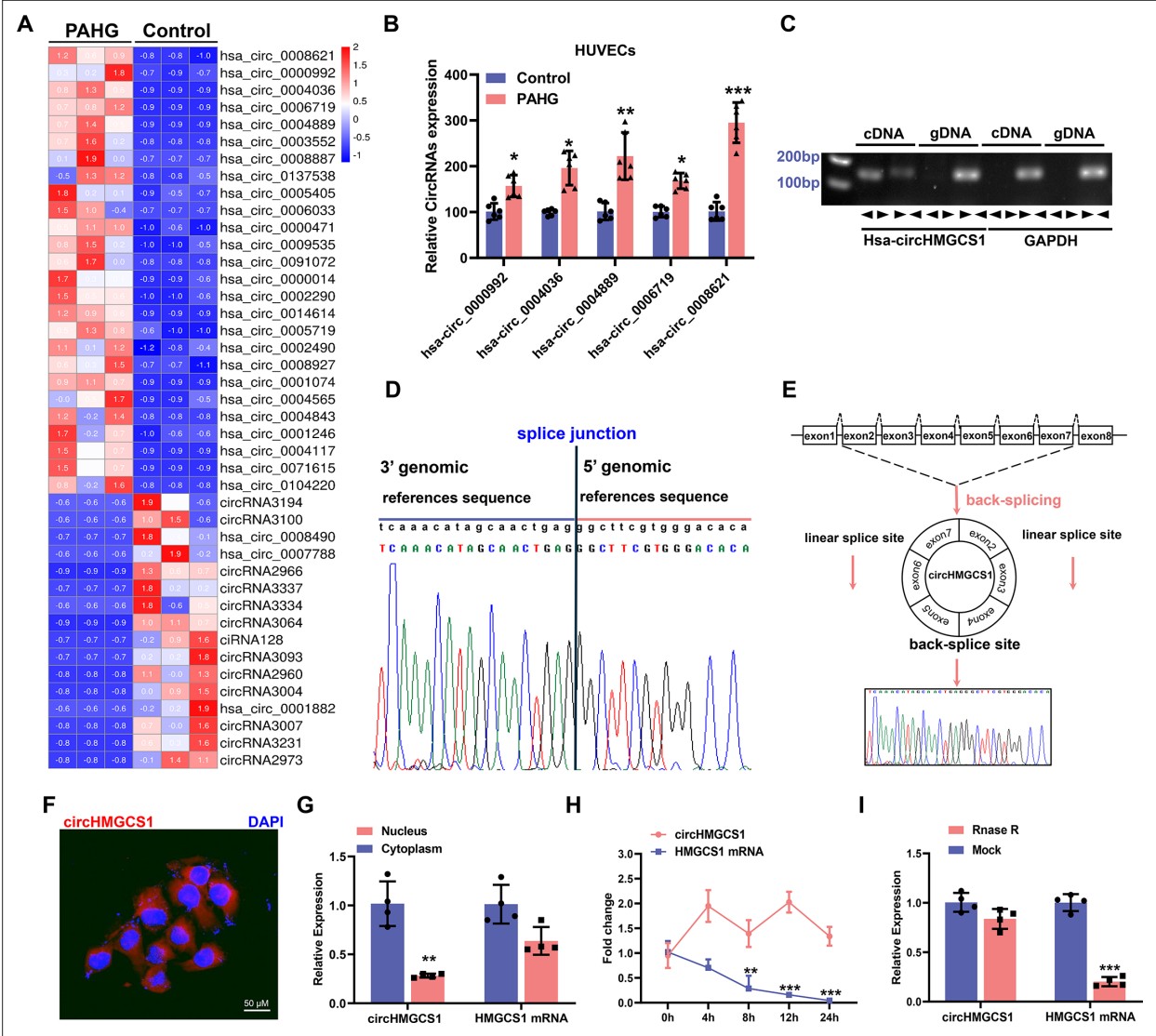

**Figure 1.** *circHMGCS1* upregulation and its association with PAHG-induced endothelial dysfunction. (**A**) Heat map illustrating the differential expression of circRNAs in HUVECs treated with PAHG for 24 hours (n=3). (**B**) qRT-PCR validation of five circRNAs in HUVECs treated with PAHG compared to normal cells, normalized to GAPDH (n=6). (**C**) PCR validation of *circHMGCS1* presence in HUVECs, *GAPDH* was utilized as a negative control (n=6). (**D**) Alignment of *circHMGCS1* sequence in CircBase (circular RNA database, upper) in agreement with Sanger sequencing results (lower). (**E**) Schematic representation of circularization of exons 2–7 of *HMGCS1* (red arrow). (**F**) RNA-FISH analysis detecting *circHMGCS1* expression in HUVECs using Cy3-labeled probes, Nuclei were counterstained with DAPI (red represents *circHMGCS1*, blue represents nucleus, scale bar=50 μm, n=4). (**G**) qRT-PCR quantification of *circHMGCS1* and *HMGCS1* mRNA levels in the cytoplasm or nucleus of HUVECs. *circHMGCS1* and *HMGCS1* mRNA levels were normalized to cytoplasmic values (n=4). (**H**) qRT-PCR measure *circHMGCS1* and *HMGCS1* mRNA levels after actinomycin D treatment (n=4 at different time points). (**I**) *circHMGCS1* expression was detected in RNase R-treated total RNA by qRT-PCR (n=4). *p<0.05, **p<0.01, ***p<0.001, All significant difference was determined by unpaired two-tailed Student's t-test or one-way ANOVA followed by Bonferroni multiple comparison post hoc test, error bar indicates SD.

The online version of this article includes the following source data and figure supplement(s) for figure 1:

**Source data 1.** Uncropped and labeled gels for (*Figure 1*).

**Source data 2.** Raw unedited gels for (*Figure 1*).

**Figure supplement 1.** Distinct expression profiles of circRNAs in endothelial dysfunction of HUVECs.

**Figure supplement 1—source data 1.** Uncropped and labeled gels for (*Figure 1—figure supplement 1*).

**Figure supplement 1—source data 2.** Raw unedited gels for (*Figure 1—figure supplement 1*).

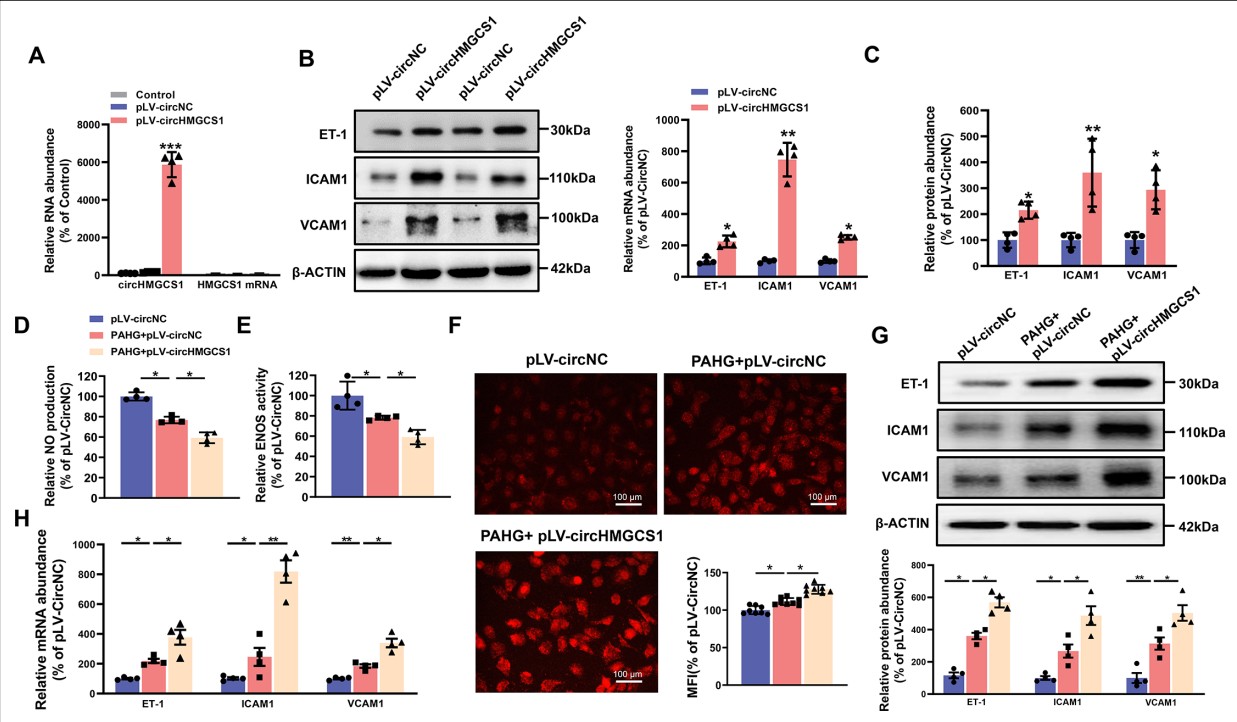

**Figure 2.** circHMGCS1 overexpression aggravates PAHG-induced endothelial dysfunction. (**A**) HUVECs were transfected with either *circHMGCS1* overexpression lentivirus (pLV-*circHMGCS1*) or lentiviral circular RNA negative control vector (pLV-circNC). After 48 hours of infection, *circHMGCS1* and *HMGCS1* expression levels were assessed by qRT-PCR and normalized to *GAPDH* (n=4). (**B, C**) Western blot and qRT-PCR were conducted to detect the expressions of adhesion molecules (*VCAM1, ICAM1,* and *ET-1*) in HUVECs between pLV-*circHMGCS1* and pLV-circNC groups (n=4). (**D**) NO content in pLV-circHMGCS1-infected HUVECs after PAHG treatment (n=4). (**E**) *ENOS* activity in pLV-circHMGCS1-infused HUVECs after PAHG treatment (n=4). (**F**) DHE fluorescence was used to characterize ROS expression in different groups. Scale bar=100 μm (n=8). (**G, H**) Relative expression of adhesion molecules (ICAM1, VCAM1 and ET-1) in HUVECs infected with pLV-circNC or pLV-*circHMGCS1* after PAHG treatment were determined by Western blot and qRT-PCR (n=4). *p<0.05, **p0.01, ***p<0.001, All significant difference was determined by one-way ANOVA followed by Bonferroni multiple comparison post hoc test, error bar indicates SD.

The online version of this article includes the following source data and figure supplement(s) for figure 2:

**Source data 1.** Uncropped and labeled gels for (*Figure 2*).

**Source data 2.** Raw unedited gels for (*Figure 2*).

**Figure supplement 1.** A sketch map of *circHMGCS1* plasmid construction.

findings suggest that elevated expression of *circHMGCS1* may contribute to the development of endothelial cell dysfunction.

## *MIR4521* protects endothelial cells from PAHG-induced dysfunction

circRNAs may function as ceRNAs to sponge miRNAs, thereby modulating miRNA target depression and imposing an additional level of post-transcriptional regulation in human diseases (*Zhong et al., 2018*). Accordingly, we conducted miRNA sequencing in PAHG-stimulated HUVECs to identify the potential miRNA targets of *circHMGCS1*. The deep sequencing analysis revealed 98 significantly differentially expressed miRNAs (*Figure 3A and B*, GEO submission: GSE237295). We then used the miRanda database and ceRNA theory to obtain four candidate miRNAs (*MIR4521, MIR3143, MIR98-5P,* and *MIR181A-2–3* P; *Figure 3C and D*). qRT-PCR confirmed that all four miRNAs were downregulated (*Figure 3E*). Next, we observed that each of the four miRNA mimics induced a significantly increase in the expression of their corresponding miRNAs in HUVECs through the application of synthetic miRNA mimics (*Figure 3—figure supplement 1A*). Relative to the other three miRNAs, *MIR4521* mimics significantly inhibited adhesion molecules (*ICAM1, VCAM1,* and *ET-1*) expression in HUVECs (*Figure 3F and G*), making it a suitable candidate for further endothelial function studies. We then investigate the effect of *MIR4521* on PAHG-stimulated endothelial cell function using synthetic *MIR4521* mimics and *MIR4521* inhibitor in HUVECs (*Figure 3—figure supplement 1B, C*). Enhanced

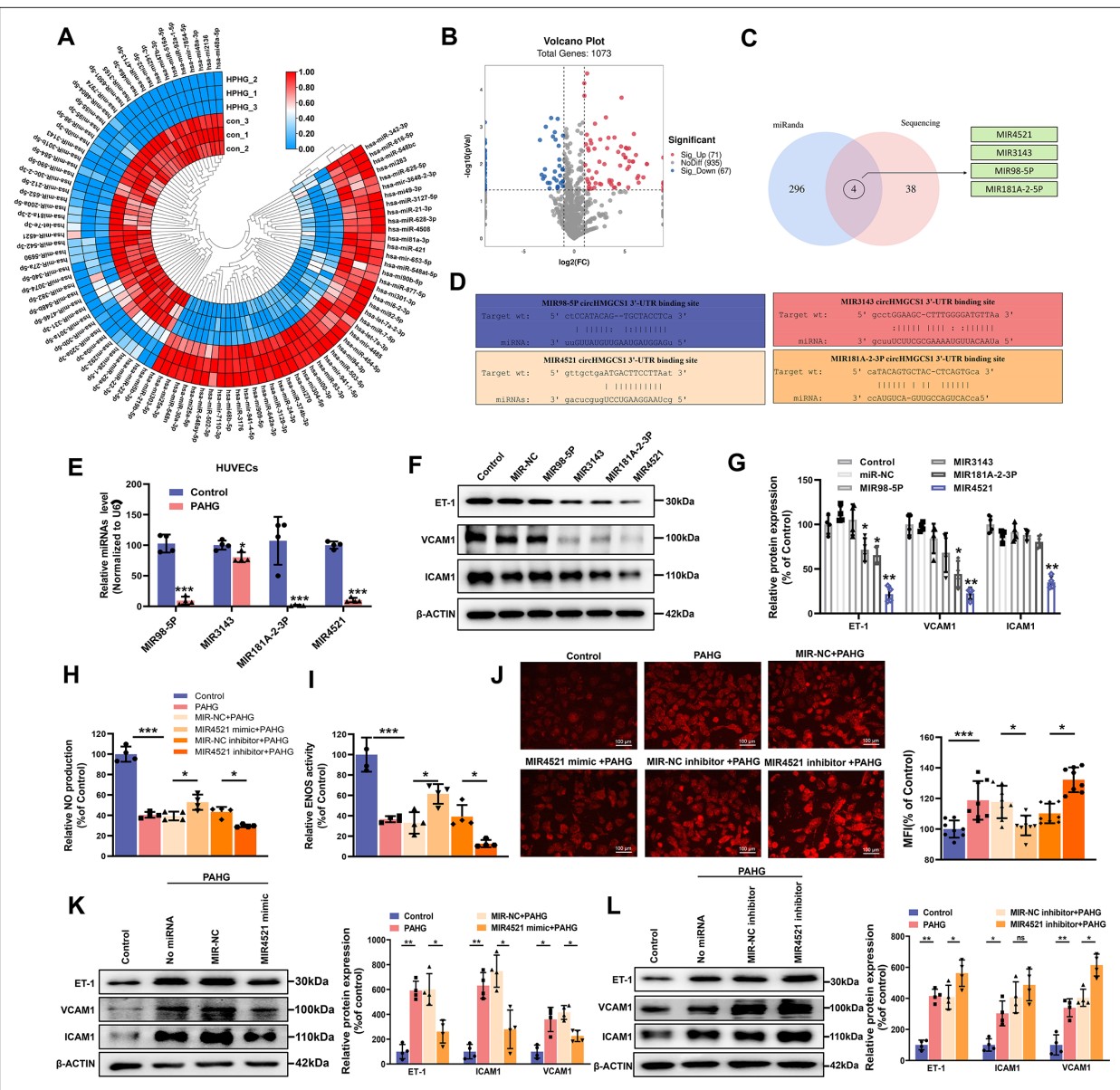

**Figure 3.** *MIR4521* prevents PAHG-induced endothelial dysfunction. (**A**) Heat map depicting differentially expressed miRNAs in HUVECs with or without PAHG treatment for 24 hr (n=3). (**B**) Volcano plot illustrating significant changes in miRNAs between control and PAHG-treated groups (Fold-change>2 or<0.5, p≤0.01). (**C**) Schematic illustration showing overlapping target miRNAs of circHMGCS1 predicated by miRanda and sequencing results. (**D**) The binding sites of *circHMGCS1* were predicted using miRanda and involve *MIR98-5P, MIR3143, MIR4521,* and *MIR181A-2–3* P. (**E**) Relative expression of four miRNA candidates in HUVECs treated with PAHG was assessed using qRT-PCR (n=4). (**F, G**) The regulatory effects of four miRNA mimics on the protein expression levels of adhesion molecules (*ICAM1, VCAM1,* and *ET-1*) (n=4). (**H**) Regulation of NO content by *MIR4521* mimic or *MIR4521* inhibitor under PAHG treatment (n=4). (**I**) Impact of *MIR4521* mimic or *MIR4521* inhibitor on *ENOS* activity during PAHG treatment. (n=4). (**J**) The ROS expression in *MIR4521* mimic or *MIR4521* inhibitor combined with PAHG treatment (Scale bar=100 μm, n=8). (**K**) Adhesion molecules (*ICAM1, VCAM1,* and *ET-1*) expression in PAHG-treated HUVECs transfected with *MIR4521* mimics was determined by Western blot (n=4). (**L**) Determination of adhesion molecules (*ICAM1, VCAM1,* and *ET-1*) expression via Western blot in PAHG-treated HUVECs transfected with *MIR4521* inhibitor (n=4). *p<0.05, **p<0.01, ***p<0.001, All significant difference was determined by one-way ANOVA followed by Bonferroni multiple comparison post hoc test, error bar indicates SD.

The online version of this article includes the following source data and figure supplement(s) for figure 3:

**Source data 1.** Uncropped and labeled gels for (*Figure 3*).

**Source data 2.** Raw unedited gels for (*Figure 3*).

**Figure supplement 1.** *MIR4521* inhibition aggravates diabetes-induced endothelial dysfunction.

*MIR4521* levels effectively restored PAHG-induced reductions in NO content and *ENOS* activity in HUEVCs, while *MIR4521* inhibition led to opposite results (*Figure 3H, I*). whereas *MIR4521* mimics significantly inhibited PAHG-induced expression of ROS and adhesion molecules (*VCAM1*, *ICAM1*, and *ET-1*; *Figure 3J*), whereas the *MIR4521* inhibitor increased the PAHG-induced expression of ROS and adhesion molecules (*VCAM1*, *ICAM1*, and *ET-1*; *Figure 3K and L* and *Figure 3—figure supplement 1D, E*). These findings suggest that *MIR4521* is involved in PAHG-induced endothelial cell dysfunction.

## *circHMGCS1* acts as a *MIR4521* sponge to regulate endothelial dysfunction

We then explored the interaction mechanism between *circHMGCS1* and *MIR4521*. As displayed in *Figure 4—figure supplement 1A, B*, the overexpression of *circHMGCS1* reduced the expression level of *MIR4521*, whereas knockdown of *circHMGCS1* resulted in an upregulation of *MIR4521*. Meanwhile, bioinformatics analysis revealed the predicted binding sites between *MIR4521* and *circHMGCS1* (*Figure 4—figure supplement 1C*). We constructed a dual-luciferase reporter by inserting the wild-type (WT) or mutant (MUT) linear sequence of circHMGCS1 into the pmirGLO luciferase vector. The co-transfection of *MIR4521* mimics with the *circHMGCS1* luciferase reporter gene in HEK293T cells, reducing luciferase activity; mutations in the binding sites between *circHMGCS1* and *MIR4521* abrogated the effect of *MIR4521* mimics on the activity of the *circHMGCS1* luciferase reporter gene mutant (*Figure 4A*). Furthermore, AGO2 immunoprecipitation in HUVECs transfected with *MIR4521* or its mutants demonstrated that *MIR4521* facilitates the association of *AGO2* with *circHMGCS1* in HUVECs (*Figure 4B*). Meanwhile, we used biotinylated *MIR4521* mimics in HUVECs stably overexpressing *circHMGCS1* to further validate the direct binding of *MIR4521* and *circHMGCS1*. The qRT-PCR results revealed that *MIR4521* captured endogenous *circHMGCS1*, and the negative control with a disrupting putative binding sequence failed to coprecipitate *circHMGCS1* (*Figure 4C*). Consistently, the use of biotin-labeled *circHMGCS1* effectively captured both *MIR4521* and *AGO2* (*Figure 4—figure supplement 1D, E*). Moreover, the RNA-FISH assay showed colocalization of *MIR4521* and *circHMGCS1* in the cytoplasm (*Figure 4D*). Altogether, these findings indicate that *circHMGCS1* functions as a molecular sponge for *MIR4521*.

Rescue experiments were conducted to investigate whether *circHMGCS1* regulates endothelial cell function through *MIR4521* sponging. We achieved *MIR4521* inhibition by transfecting HUVECs with a recombinant adeno-associated virus 9 (AAV9) vector carrying *MIR4521* sponges. Notably, *MIR4521* sponges significantly promoted the PAHG-induced reduction in NO content and ENOS activity. Under this condition, increased circHMGCS1 expression did not interfere with the changes in NO content and *ENOS* activity (*Figure 4E and F*). Moreover, *MIR4521* sponges elevated ROS expression in PAHG treated HUVECs, whereas no further additional effect on ROS expression was observed upon the overexpression of *circHMGCS1* when *MIR4521* was sponged (*Figure 4G*). Further evaluation of endothelial functional molecules revealed that circHMGCS1 failed to stimulate the expression of these factors when *MIR4521* was sponged (*Figure 4H–J*). Furthermore, the reduction in NO content and *ENOS* activity induced by *circHMGCS1* overexpression was reversed by *MIR4521* mimics (*Figure 4K and L*), and the *circHMGCS1*-mediated increase in ROS content was inhibited by *MIR4521* mimics (*Figure 4M*). Furthermore, the expression of adhesion molecules (*ICAM1*, *VCAM1*, and *ET-1*) was increased in *circHMGCS1*-transfected HUVECs, and this effect was significantly ameliorated by *MIR4521* mimics (*Figure 4N–P*). These findings demonstrate that *circHMGCS1* acts as a sponge for *MIR4521*, thereby regulating endothelial function.

## *circHMGCS1* serves as a *MIR4521* sponge to regulate diabetes-induced VED

To further explore the interaction between *MIR4521* and *circHMGCS1* in diabetes-induced VED, we administered exogenous *MIR4521* agomir (mimics) via tail vein injection to mice, whereas agomir NC, expressing random sequence, served as the negative control in HFHG-induced diabetes to examine its effect on diabetes and associated VED (*Figure 5—figure supplement 1A*). Compared with to non-diabetic WT (Control) mice, DM mice exhibited a significant increase in body weight, upon *MIR4521* agomir application, the body weight of diabetic mice significantly decreased compared with the untreated diabetic mice (*Figure 5A*), and elevated fasting blood glucose levels in DM mice were likewise substantially inhibited (*Figure 5B*). By contrast, the combined treatment of *circHMGCS1* and

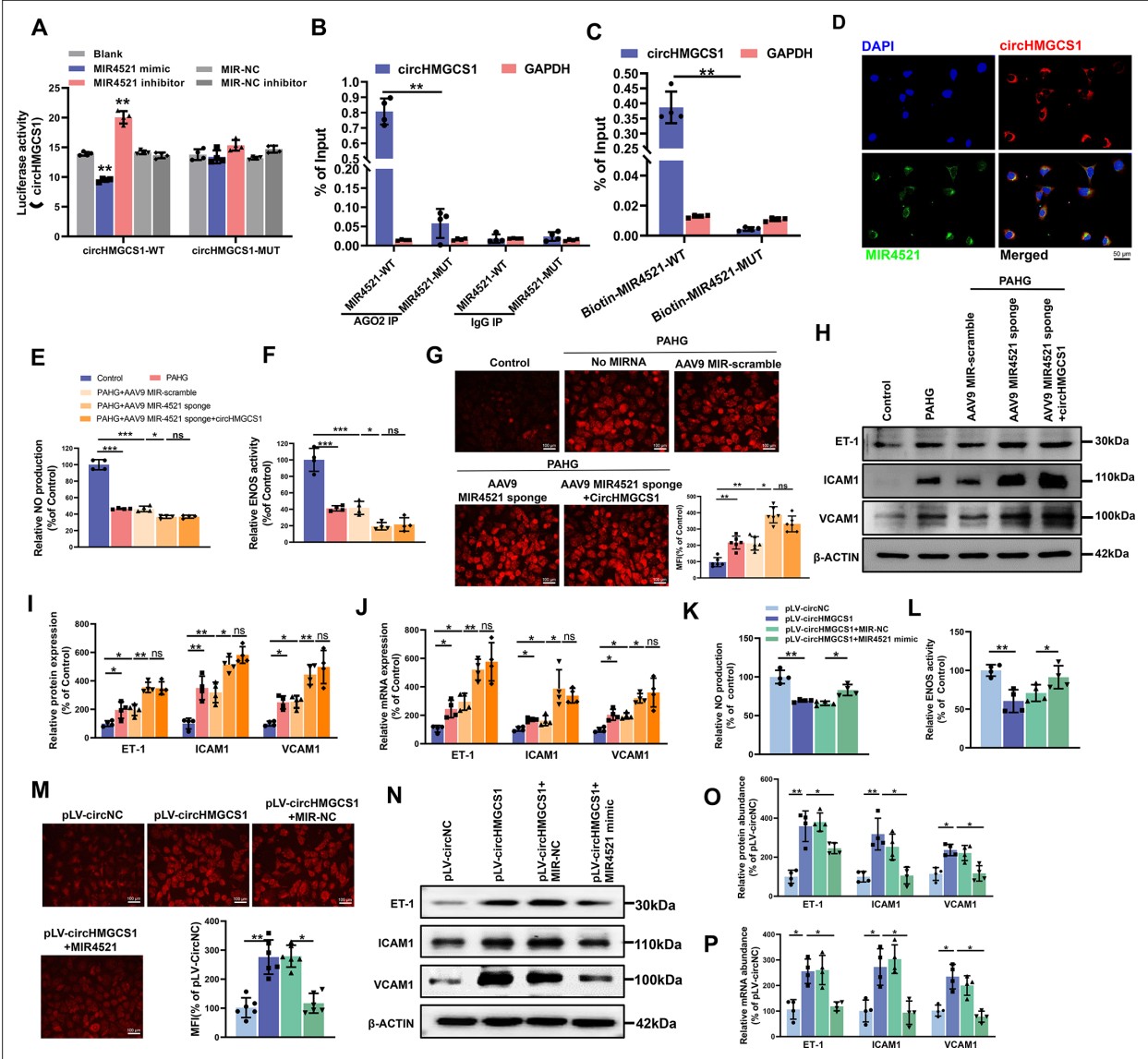

**Figure 4.** *circHMGCS1* regulates PAHG-induced endothelial dysfunction by targeting and sponging *MIR4521*. (**A**) Luciferase reporter constructs containing wild-type (WT) or mutant (MUT) circHMGCS1 were cotransfected with *MIR4521* mimics, MIR-NC, *MIR4521* inhibitor, or MIR-NC inhibitor in HEK293T cells (n=4). (**B**) Immunoprecipitation shows *AGO2*-mediated binding of *circHMGCS1* and *MIR4521* (n=4). (**C**) Biotin-coupled *MIR4521* or its mutant probe was employed for *circHMGCS1* pull-down, and captured *circHMGCS1* level was quantified by qRT-PCR (n=4). (**D**) RNA-FISH showing the colocalization of *circHMGCS1* and *MIR4521* in HUVECs (red represents *circHMGCS1*, green represents *MIR4521*, blue represents nucleus, scale bar=50 μm, n=4). (**E, F**) *circHMGCS1* exhibited no impact on NO content and *ENOS* activity in the presence of *MIR4521* sponge (n=4). (**G**) *circHMGCS1* demonstrated no influence on ROS expression with *MIR4521* sponge (Scale bar=100 μm, n=8). (**H–J**) *circHMGCS1* had no effect on the expression of adhesion molecules (*ICAM1*, *VCAM1*, and *ET-1*) in the presence of *MIR4521* sponge which were determined by western blot and qRT-PCR (n=4). (**K, L**) *MIR4521* attenuated the reduction of NO content and *ENOS* activity induced by *circHMGCS1* (n=4). (**M**) *MIR4521* inhibited the increase of ROS expression caused by *circHMGCS1* (Scale bar=100 μm, n=8). (**N–P**) *MIR4521* attenuated the increased expression of adhesion molecules (*ICAM1*, *VCAM1*, and *ET-1*) induced by *circHMGCS1* which were determined by Western blot and qRT-PCR, respectively (n=4). *p<0.05, **p<0.01, ns means no significant. All significant difference was determined by one-way ANOVA followed by Bonferroni multiple comparison post hoc test, error bar indicates SD.

The online version of this article includes the following source data and figure supplement(s) for figure 4:

**Source data 1.** Uncropped and labeled gels for (***Figure 4***).

**Source data 2.** Raw unedited gels for (***Figure 4***).

**Figure supplement 1.** Verification of the binding site of *MIR4521* and *circHMGCS1*.

**Figure supplement 1—source data 1.** Uncropped and labeled gels for (***Figure 4—figure supplement 1***).

**Figure supplement 1—source data 2.** Raw unedited gels for (***Figure 4—figure supplement 1***).

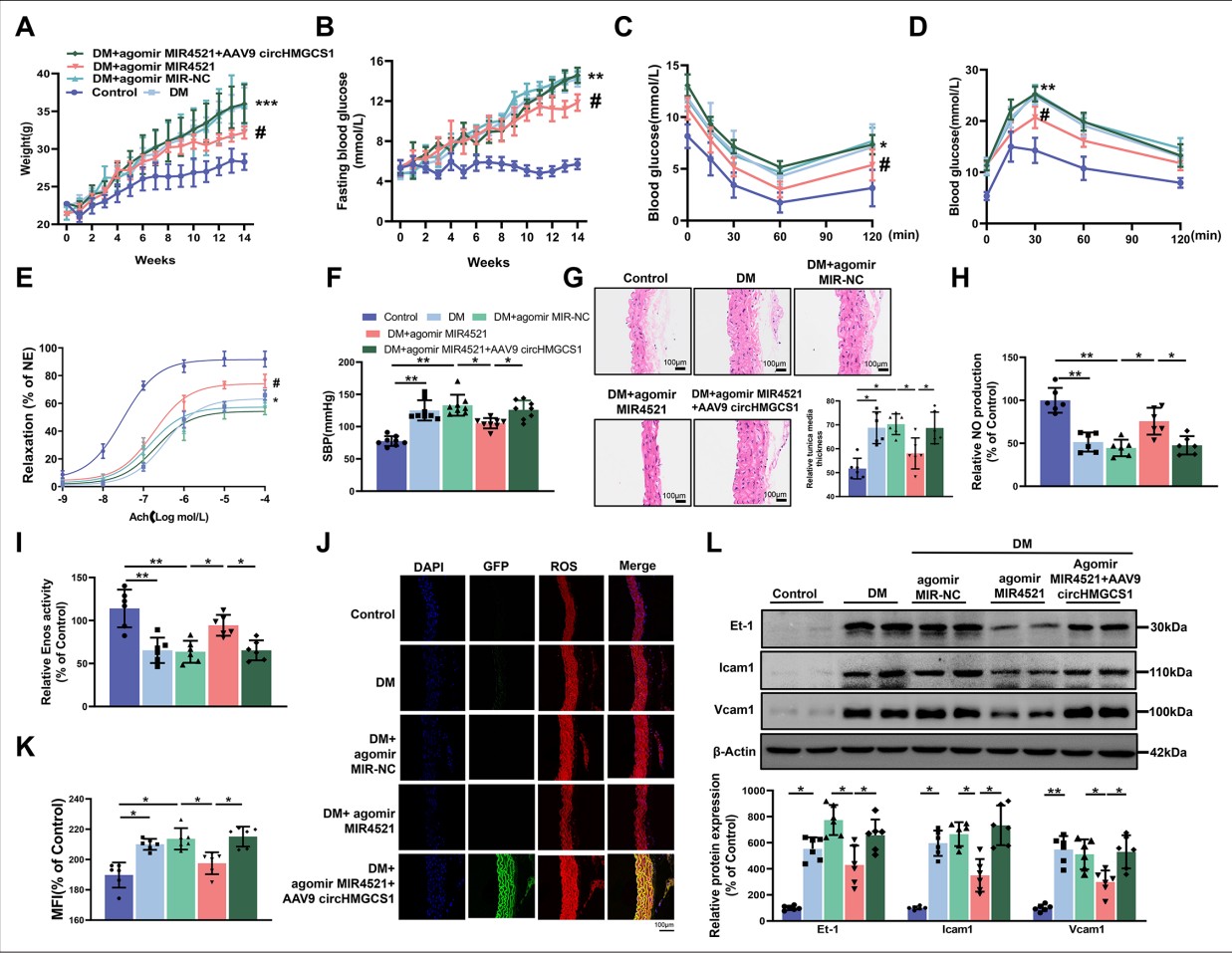

**Figure 5.** AAV9-mediated *circHMGCS1* overexpression attenuates the protective effect of *MIR4521* against diabetes-induced VED. (**A**) Changes in body weight of mice from 0 to 14 weeks (n=8, ***p<0.001 DM versus control, #p<0.05 DM+agomir MIR4521 versus DM). (**B**) Alterations in fasting blood glucose in mice from 0 to 14 weeks (n=8, ***p<0.001 DM versus control, #p<0.05 DM+agomir *MIR4521* versus DM). (**C, D**) ITT and OGTT were performed in the fasted mice (n=8, *p<0.05 DM versus control, #p<0.05 DM+agomir *MIR4521* versus DM). (**E**) Relaxation responses of aortic rings from control, *MIR4521* agomir, or *MIR4521* agomir +AAV9 *circHMGCS1* mice on DM diet (n=6). (**F**) SBP measured via tail-cuff method in different groups (n=8). (**G**) Hematoxylin and eosin staining (H&E) was performed on serial cross-sections of thoracic aortas from differently treated mice to assess vessel wall thickness. Scale bar=100 µm (n=6). (**H, I**) NO content and *Enos* activity in thoracic aorta after 6 weeks of DM diet with *MIR4521* agomir or *MIR4521* agomir +*circHMGCS1* treatment (n=6). (**J, K**) Detection of ROS expression in thoracic aorta using DHE after *MIR4521* agomir or *MIR4521* agomir +AAV9 *circHMGCS1* injection (red represents ROS, green represents GFP, blue represents nucleus, scale bar=100 µm, n=6). (**L**) Expression levels of adhesion molecules (*Icam1*, *Vcam1*, and *Et-1*) in thoracic aorta after 6 weeks of DM diet with *MIR4521* agomir or *MIR4521* agomir +*circHMGCS1* treatment (n=6). *p<0.05, **p<0.01, ***p<0.001. All significant difference was determined by one-way ANOVA followed by Bonferroni multiple comparison post hoc test, error bar indicates SD.

The online version of this article includes the following source data and figure supplement(s) for figure 5:

**Source data 1.** Uncropped and labeled gels for (*Figure 5*).

**Source data 2.** Raw unedited gels for (*Figure 5*).

**Figure supplement 1.** *circHMGCS1* counteracts the protective role of *MIR4521* in diabetes-induced VED.

*MIR4521* agomir did not significantly affect the body weight and blood glucose levels. OGTT and ITT experiments demonstrated that *MIR4521* agomir considerably enhanced glucose tolerance and insulin resistance in diabetic mice (*Figure 5C and D* and *Figure 5—figure supplement 1B, C*). Blood lipid biochemistry analysis revealed that *MIR4521* agomir restored abnormal blood lipid levels in diabetic mice (*Figure 5—figure supplement 1D–G*). Notably, the abundant presence of *circHMGCS1* in the body can negate the inhibitory effect of *MIR4521* on diabetes development. These results suggest that *MIR4521* can inhibit the occurrence of diabetes, whereas *circHMGCS1* specifically dampens the function of *MIR4521*, weakening its protective effect against diabetes.

To investigate the roles of *MIR4521* and *circHMGCS1* in mouse vascular relaxation, we conducted an ex vivo culture of thoracic aortas from mice subjected to different treatments. A significant impairment in acetylcholine (Ach)-induced vascular relaxation response in the thoracic aorta of diabetic mice was markedly improved by *MIR4521* agomir, emphasizing the critical involvement of *MIR4521* in vascular relaxation. Conversely, *circHMGCS1* inhibited the protective function of *MIR4521* on vascular relaxation (*Figure 5E*). Across all experimental groups, the smooth muscle exhibited a normal endothelium-dependent relaxation response to the nitric oxide donor sodium nitroprusside (SNP; *Figure 5—figure supplement 1H*), revealing intact smooth muscle function. Meanwhile, diabetic mice displayed a significant increase in SBP compared with normal mice, which was significantly restrained by *MIR4521* agomir treatment. Nevertheless, this beneficial effect was nullified in the presence of *circHMGCS1* (*Figure 5F*). Furthermore, *MIR4521* agomir inhibited intimal thickening and smooth muscle cell proliferation in diabetic mice, whereas upregulation of *circHMGCS1* abrogated this effect (*Figure 5G*). These findings demonstrate that *circHMGCS1* and *MIR4521* play specific roles in regulating vascular endothelial function in diabetic mice.

We measured the NO level to further investigate the interaction between *MIR4521* and *circHMGCS1* in regulating diabetes-induced VED. Compared with the control group, the NO level decreased in the thoracic aorta of diabetic mice; however, *MIR4521* agomir treatment increased its content (*Figure 5H*). Furthermore, *MIR4521* treatment effectively enhanced *Enos* activity (*Figure 5I*), whereas *circHMGCS1* hindered *MIR4521* regulatory function on NO content and *ENOS* activity. Furthermore, *MIR4521* reduced ROS production in the blood vessels (*Figure 5J and K*) and inhibited the expression of endothelial adhesion molecules (*Icam1*, *Et-1*, and *Vcam1*; *Figure 5L*). Nonetheless, in the presence of *circHMGCS1*, the effect of *MIR4521* was abrogated. These results indicated that *circHMGCS1* acts as a *MIR4521* sponge, participating in diabetes-induced VED.

## *ARG1* is a direct target of *MIR4521* to accelerate endothelial dysfunction

To investigate whether *circHMGCS1*-associated *MIR4521* regulated VED-related gene expression by targeting the 3'-untranslated region (UTR), we conducted predictions of mRNAs containing *MIR4521* binding sites using the GenCard, mirDIP, miRWalk, and Targetscan databases. Through bioinformatics analyses, we identified binding sites for *MIR4521* on *KLF6*, *ARG1*, and *MAPK1* genes (*Figure 6A*). Previous studies have demonstrated the involvement of elevated arginase activity in diabetes-induced VED. Arginase can modulate the production of NO synthesized by *ENOS* through competitive utilization of the shared substrate L-arginine (*Jung et al., 2010*; *Romero et al., 2008*). Moreover, *MIR4521* possesses the capability to bind to the 3' untranslated region of *ARG1* in both human and mouse genomes (*Figure 6—figure supplement 1A*). The protein expression of *Arg1* was increased in the aorta of diabetic mice and PAHG-treated HUVECs (*Figure 6B* and *Figure 6—figure supplement 1B*). The luciferase reporter assay was applied to verify the targeting ability through the pmirGLO vector, which included either the WT or MUT 3'-UTR of *ARG1* (*Figure 6—figure supplement 1C*). The overexpression of *MIR4521* reduced the luciferase activities of the WT reporter vector but not the MUT reporter vector (*Figure 6C*). We next investigated the function of *MIR4521* and *ARG1* to regulate the function of HUVECs. *MIR4521* mimics recovered the reduced levels of NO and the compromised activity of *ENOS* triggered by *ARG1* overexpression (*Figure 6D and E* and *Figure 6—figure supplement 1D, E*). Meanwhile, it inhibited the upregulation in ROS levels induced by *ARG1* overexpression (*Figure 6F* and *Figure 6—figure supplement 1F, G*). The rescue experiment results demonstrated that *MIR4521* mimics reversed the promotional effect of *ARG1* on the mRNA and protein expression of adhesion molecules in vitro (*Figure 6G and H*). Subsequently, we evaluated the gene and protein expression levels of ARG1 using qRT-PCR and western blotting. *MIR4521* mimics repressed *ARG1* transcription and protein expression, whereas MIR4521 inhibition upregulated *ARG1* expression (*Figure 6I–K*). Taken together, these results suggest that *MIR4521* inhibits VED through sponging *ARG1*.

To explore the regulatory effect of the interaction between *circHMGCS1* and *MIR4521* on *ARG1* expression. we firstly used HUVECs employing overexpression of *circHMGCS1*. *circHMGCS1* overexpression significantly further promoted *ARG1* transcription and protein expression during the PAHG treatment (*Figure 6L and M*), confirming the regulatory function of *circHMGCS1* in VED-related *ARG1* expression. We next studied the function of *MIR4521*, which acts as a sponge for *circHMGCS1*

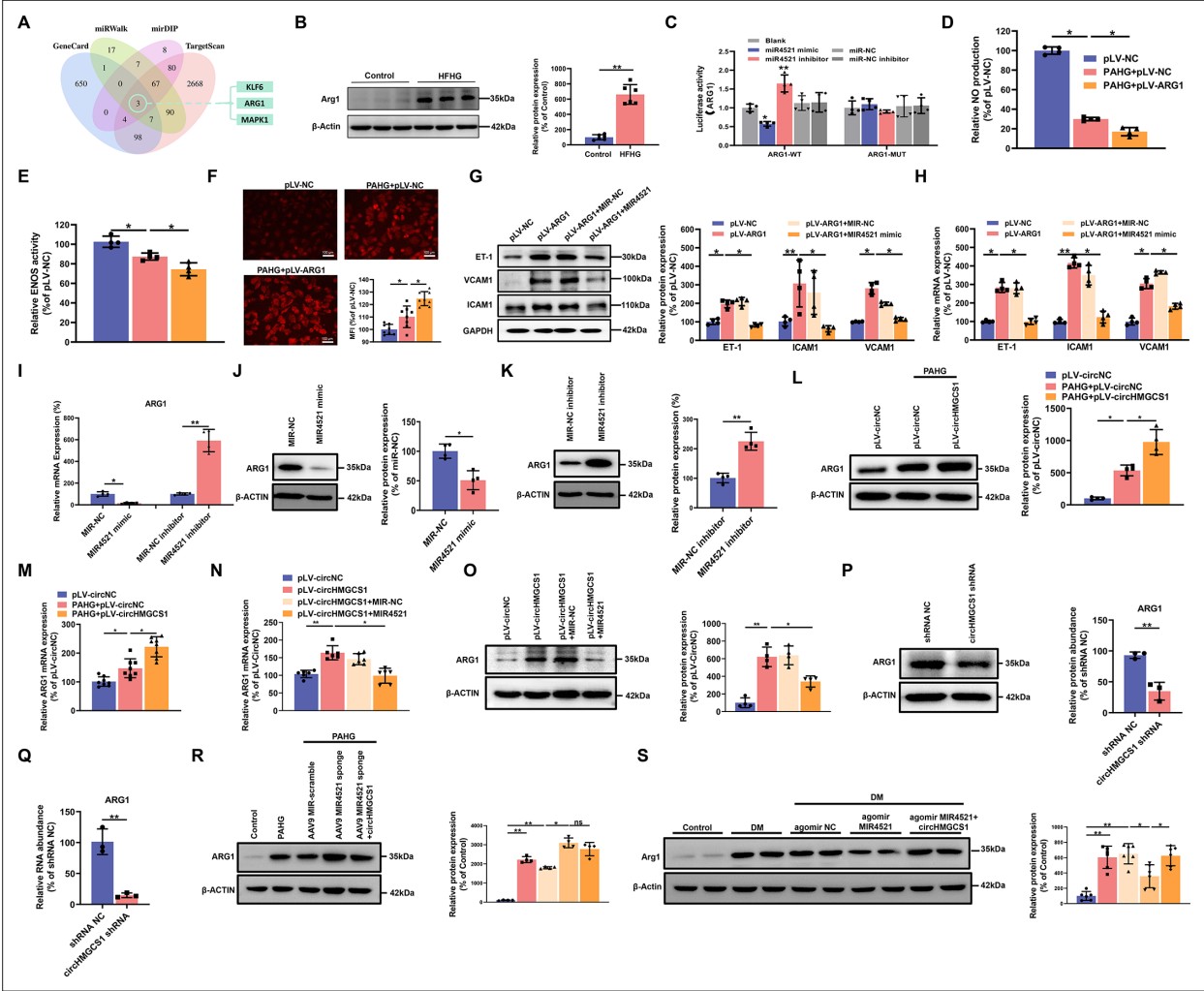

**Figure 6.** *circHMGCS1* functions as a *MIR4521* sponge in HUVECs to modulate *ARG1* expression. (**A**) Schematic depiction illustrating the intersection of predicted target genes of *MIR4521* from TargetScan, miRWalk, mirDIP, and GeneCard. (**B**) *Arg1* expression in thoracic aorta of DM by western blot (n=6). (**C**) HEK293T cells were cotransfected with *MIR4521* mimics, MIR-NC, *MIR4521* inhibitor, or MIR-NC inhibitor, along with luciferase reporter constructs containing WT or MUT 3'-untranslated region of *ARG1* (n=4). (**D, E**) Modulatory effect of *MIR4521* mimic on NO content and *ENOS* activity under ARG1 overexpression (n=4). (**F**) Regulatory impact of *MIR4521* mimic on ROS content under *ARG1* overexpression (Scale bar=100 μm, n=4). (**G, H**) *ARG1*-overexpressing HUVECs were generated using lentivirus and transfected with *MIR4521* mimics for 24 hr to evaluate adhesion molecule expression (*ICAM1*, *VCAM1*, and *ET-1*) by western blot and qRT-PCR (n=4). (**I–K**) *ARG1* expression was significantly decreased by *MIR4521* mimics and increased by *MIR4521* inhibitor, as determined by qRT-PCR and Western blot (n=4). (**L, M**) *ARG1* expression was significantly increased by *circHMGCS1* overexpression, as determined by western blot and qRT-PCR (n≥4). (**N, O**) *circHMGCS1*-overexpressed HUVECs, created with lentivirus and transfected with *MIR4521* mimics for 48 hr, were examined for *ARG1* expression by western blot and qRT-PCR (n=4). (**P, Q**) *ARG1* expression was significantly reduced by *circHMGCS1* shRNA, as determined by western blot and qRT-PCR (n=3). (**R**) AAV9 *MIR4521* sponge was used to inhibit the *MIR4521* expression in HUVECs, followed by *circHMGCS1* transfection, and then treated with PAHG for 24 hr to evaluate the expression of *ARG1* by western blot (n=4). (**S**) Detection of *Arg1* expression in the thoracic aorta of DM treated with AAV9 *MIR4521* agomir combined with AAV9 *circHMGCS1* by western blot (n=4). *p<0.05, **p<0.01, ns means no significant. All significant difference was determined by one-way ANOVA followed by Bonferroni multiple comparison post hoc test, error bar indicates SD.

The online version of this article includes the following source data and figure supplement(s) for figure 6:

**Source data 1.** Uncropped and labeled gels for (*Figure 6*).

**Source data 2.** Raw unedited gels for (*Figure 6*).

**Figure supplement 1.** *MIR4521* modulates the expression of *ARG1*.

**Figure supplement 1—source data 1.** Uncropped and labeled gels for (*Figure 6—figure supplement 1*).

**Figure supplement 1—source data 2.** Raw unedited gels for (*Figure 6—figure supplement 1*).

to regulate *ARG1* in HUVECs. The mRNA and protein levels of *ARG1* were significantly increased when HUVECs were transfected with *circHMGCS1*. However, *MIR4521* overexpression counteracted the effects of *circHMGCS1* on *ARG1* mRNA and protein expression (*Figure 6N and O*). Conversely, knockdown of *circHMGCS1* reduced *ARG1* expression and increased *MIR4521* expression in HUVECs (*Figure 6P and Q* and *Figure 4—figure supplement 1B*). Besides, the inhibition of *MIR4521* using a *MIR4521* sponge in HUVECs further increased *ARG1* expression under the PAHG treatment. Notably, elevated *circHMGCS1* expression did not affect *ARG1* regulation (*Figure 6R*). Moreover, compared with diabetic mice, *circHMGCS1* overexpression dampened the function of *MIR4521* agomir on *Arg1* (*Figure 6S*). These findings indicate that *circHMGCS1* serves as a sponge for *MIR4521*, facilitating *ARG1* regulation and VED promotion.

### *ARG1* is essential for *circHMGCS1* and *MIR4521* to regulate diabetes-induced VED

To delve further into how *circHMGCS1* and *MIR4521* regulated endothelial cell function via *ARG1*, we used AAV9 expressing *ARG1* shRNA (*ARG1* shRNA) or GFP (NC shRNA, used as the control) to reduce *ARG1* levels. *ARG1* shRNA mitigated the effects of PAHG on NO content and *ENOS* activity (*Figure 7A and B*), and inhibited PAHG-induced ROS in endothelial cells (*Figure 7C*). Additionally, *ARG1* shRNA significantly reduced the elevated expression of adhesion molecules (*ICAM1*, *VCAM1*, and *ET-1*) induced by PAHG (*Figure 7D and E*). F Further rescue experiments revealed that *MIR4521* and *circHMGCS1* exhibited no significant regulatory effect on endothelial function in the presence of ARG1 shRNA. These findings demonstrate that ARG1 plays a crucial regulatory effect on *MIR4521* and *circHMGCS1* in modulating endothelial cell function.

Considering the role of *ARG1* in regulating endothelial cell function through *circHMGCS1* and *MIR4521*, we downregulated *ARG1* expression in mice by administering AAV9 *ARG1* shRNA through tail vein injection (*Figure 7—figure supplement 1A*). As anticipated, AAV9 *ARG1* shRNA-infused mice exhibited decreased body weight and lower fasting blood glucose levels compared with diabetic mice (*Figure 7F and G*). Moreover, OGTT and ITT demonstrated that reduced *ARG1* levels in AAV9 *ARG1* shRNA-treated mice preserved glucose tolerance and insulin sensitivity (*Figure 7H, I* and *Figure 7—figure supplement 1B, C*). The lowered *ARG1* levels also reversed serum lipid levels (*Figure 7—figure supplement 1D–G*). However, the overexpression of *circHMGCS1* or *MIR4521* had no significant effect on glucose and lipid metabolism in AAV9 *ARG1* shRNA-expressing mice. These findings confirm that *ARG1* is an indispensable regulator for *circHMGCS1* and *MIR4521* in regulating diabetes.

We next investigated whether *ARG1* regulates vascular endothelial cell function in the context of diabetes. Knocking down *ARG1* in the thoracic aorta of mice expressing AAV9 *ARG1* shRNA restored diastolic function of the thoracic aorta in diabetic mice (*Figure 7J*), emphasizing the vital role of *ARG1* in VED. Furthermore, all experimental groups exhibited a normal endothelium-dependent relaxation response to the NO donor SNP, suggesting intact smooth muscle function (*Figure 7—figure supplement 1H*). Moreover, AAV9 *ARG1* shRNA effectively inhibited diabetes-induced intima thickening and inhibited smooth muscle cell proliferation (*Figure 7K*), and which also counteracted the elevation of SBP induced by diabetes (*Figure 7L*). However, the regulatory effects of *circHMGCS1* and *MIR4521* on vascular dilation were ineffective in the presence of *ARG1* shRNA. We next measured VED-related functional factors. Compared with diabetic mice, the thoracic aorta of mice expressing AAV9 *ARG1* shRNA exhibited restored NO content and *Enos* activity (*Figure 7M and N*). Furthermore, knocking down *ARG1* reduced the high expression levels of vascular endothelial adhesion molecules (*Icam1*, *Et-1*, and *Vcam1*) induced by diabetes (*Figure 7O*). Immunofluorescence analysis demonstrated a significant decrease in ROS content in the thoracic aorta of mice expressing AAV9 *ARG1* shRNA compared with diabetic mice (*Figure 7P and Q*). Notably, *circHMGCS1* and *MIR4521* lost their regulatory capacity on vascular endothelial cell function in the presence of *ARG1* shRNA. These results indicate that *ARG1* plays a critical role in *circHMGCS1* and *MIR4521* to modulate diabetes-induced VED.

## Discussion

The field of ncRNA biology has garnered considerable attention and has witnessed intensive research over the last few years. ncRNAs possess the capacity to function as novel regulators in various

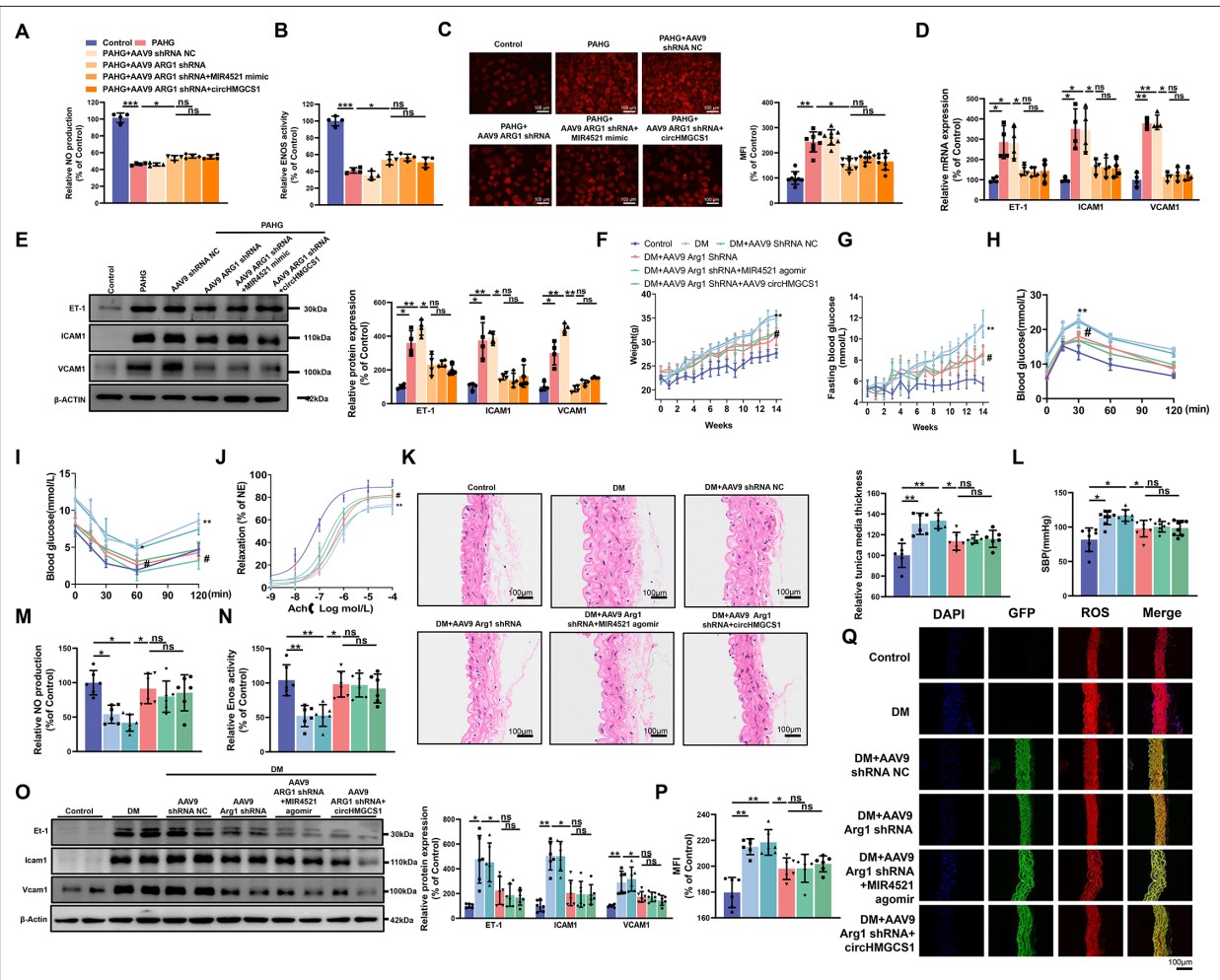

**Figure 7.** *ARG1* is inseparable from *circHMGCS1* and *MIR4521* regulating diabetes-induced VED. (**A, B**) *circHMGCS1* and *MIR4521* had no effect on NO content and *ENOS* activity expression in the absence of *ARG1* (n=4). (**C**) ROS expression remained unaffected by *circHMGCS1* and *MIR4521* in the absence of *ARG1* (n=4). (**D, E**) Relative expression of adhesion molecules (*ICAM1*, *VCAM1*, and *ET-1*) remained unchanged in the absence of *ARG1*, despite the presence of *circHMGCS1* and *MIR4521*, as determined by qRT-PCR and Western blot (n=4). (**F**) Changes in mice body weight over the experimental period (n=8, **p<0.01, DM versus control, #p<0.05, DM +AAV9 *ARG1* shRNA versus DM). (**G**) Fasting blood glucose levels in mice over time (n=8, **p<0.01, DM versus control, #p<0.05, DM +AAV9 *ARG*1 shRNA versus DM). (**H, I**) Blood glucose levels measured at week 13 (ITT) and week 14 (OGTT) (n=8, **p<0.01 DM versus control, #p<0.05, DM +AAV9 *ARG1* shRNA versus DM). (**J**) Endothelium-dependent relaxations in aortic rings from different groups (n=8). (**K**) H&E performed on serial cross-sections of thoracic aortas from differently treated mice to evaluate vessel wall thickness. Scale bar=100 μm (n=6). (**L**) SBP measured by the tail-cuff method in different groups (n=8). (**M, N**) NO content and *Enos* activity expression in thoracic aort (n=6). (**O**) Relative expression of adhesion molecules (*Icam1*, *Vcam1*, and *Et-1*) in thoracic aorta assessed by Western blot (n=6). (**P, Q**) ROS expression in the thoracic aorta using the DHE probe (Red represents ROS, Green represents GFP, Blue represents nucleus, scale bar=100 μm, n=6). *p<0.05, **p<0.01, ns means no significant. All significant difference was determined by one-way ANOVA followed by Bonferroni multiple comparison post hoc test, error bar indicates SD.

The online version of this article includes the following source data and figure supplement(s) for figure 7:

**Source data 1.** Uncropped and labeled gels for (*Figure 7*).

**Source data 2.** Raw unedited gels for (*Figure 7*).

**Figure supplement 1.** Loss of functional regulation of *circHMGCS1* and *MIR4521* on HUVECs in the absence of *ARG1* in vivo.

physiological systems and disease contexts (*Esteller, 2011*; *Matsui and Corey, 2017*; *Statello et al., 2021*). However, most studies have explored the expression patterns and functions of ncRNAs within single disease settings (*Arcinas et al., 2019*; *Shan et al., 2017*). The present study contributes to the understanding of ncRNAs and their involvement in diabetes-associated CVD, characterized by disruption of lipid metabolic homeostasis and redox balance in vascular endothelial cells under diabetic conditions. A comprehensive genome-wide analysis revealed 17,179 known circRNA loci transcribed

in diabetic endothelial cells. Based on conservation, endothelial specificity, and abundance criteria, five circRNAs were selected for validation, indicating their predominant upregulation in the diabetic endothelial environment. Among them, we discovered a novel circRNA, *circHMGCS1*, exhibiting high abundance, stability, and cell specificity in diabetic endothelial cells and contributing to endothelial function regulation. Our findings establish that *circHMGCS1* promotes VED progression, and its overexpression exacerbates endothelial cell dysfunction. Our study highlights the pivotal role of *circHMGCS1* in regulating endothelial cell function during diabetes.

circRNAs play regulatory roles by acting as microRNA sponges or interacting with proteins to regulate alternative splicing, transcription, and epigenetic modifications (*Arcinas et al., 2019*; *Hansen et al., 2013*; *Memczak et al., 2013*). To investigate the mechanism of circHMGCS1 in VED regulation, we analyzed miRNA sequencing results and prediction software to identify *MIR4521* as a novel endothelial-expressed miRNA. *MIR4521* was found to possess a stable binding site with *circHMGCS1*, and its expression was significantly decreased in endothelial cells within a diabetic environment. Our functional studies revealed that *MIR4521* mimics attenuated VED development in diabetes, whereas its inhibition exacerbated VED progression. These findings preliminarily demonstrate the regulatory ability of *MIR4521* in VED. The interaction between *circHMGCS1* and *MIR4521* was further characterized through luciferase reporter gene assays, RNA pull-down, and RNA immunoprecipitation. AAV9-mediated *circHMGCS1* overexpression in endothelial cells dampened *MIR4521* expression and accelerated diabetes-induced VED. Conversely, restoring *MIR4521* expression effectively alleviated VED progression, highlighting the critical role of *MIR4521* in counteracting *circHMGCS1*-mediated promotion of diabetes-induced VED. Moreover, the absence of *MIR4521* aggravated diabetes-induced VED, whereas exogenous *circHMGCS1* addition did not affect VED development. Intriguingly, exogenous *MIR4521* significantly restrained diabetes-induced VED exacerbation, which was attenuated upon the subsequent addition of exogenous *circHMGCS1*. These findings underscore the essential regulatory role of the *circHMGCS1* and *MIR4521* interaction in maintaining diabetic vascular homeostasis.

Finally, we investigated the downstream targets crucial for *circHMGCS1*-mediated endothelial cell function. Arginase is a dual-nucleus manganese metalloenzyme that catalyzes the hydrolysis of L-arginine, primarily producing L-ornithine and urea (*Kanyo et al., 1996*). *ARG1* and *ARG2*—two isoforms of arginase—exist in vertebrates, including mammals. Although *ARG1* and *ARG2* share the same catalytic reaction, they are encoded by different genes and exhibit distinct immunological characteristics (*Hara et al., 2020*; *Pudlo et al., 2017*; *Su et al., 2021*). Elevated *ARG1* activity in endothelial cells has been identified as a significant contributor to VED in T2DM individuals. This dysfunction is attributed to the excessive formation of ROS caused by the competition between *ARG1* and *ENOS* for L-arginine. Consequently, oxidases or mitochondrial complexes become overactivated, resulting in reduced NO levels and subsequent VED (*Caldwell et al., 2018*). By contrast, inhibiting arginase has been shown to considerably enhance endothelial function in T2DM patients (*Mahdi et al., 2018*; *Shemyakin et al., 2012*; *Zhou et al., 2018*). Our findings suggest that T2DM enhances *ARG1* activity and expression in vascular endothelial cells. Conversely, *ARG1* inhibition in endothelial cells delayed the onset of diabetes and preserved NO metabolite levels in the aorta. Additionally, Ach-induced vasodilation remains intact in the absence of *ARG1* in the aortic ring, as *ARG1* inhibition promotes *ENOS*-dependent relaxation, thereby preventing VED. Meanwhile, ROS content in the aorta of diabetic mice could be reduced under *ARG1* inhibition, thereby suppressing the occurrence of oxidative stress. AAV9-mediated overexpression of *circHMGCS1* increases *ARG1* expression, which is inhibited by exogenous *MIR4521* intervention. In the absence of *MIR4521*, *circHMGCS1* cannot regulate *ARG1* expression, whereas exogenous *MIR4521* intervention inhibits the diabetes-induced increase in *ARG1* expression in vivo, and this effect is counteracted by *circHMGCS1* overexpression. Both in vitro and in vivo data suggest that the *MIR4521* or *circHMGCS1* fails to regulate the effect of diabetes-induced VED in the absence of *ARG1*. Therefore, *ARG1* may serve as a promising VED biomarker, and *circHMGCS1* and *MIR4521* play a key role in regulating diabetes-induced VED by *ARG1*.

It would be intriguing to elucidate the mechanisms through which *circHMGCS1* and *MIR4521* exert their regulatory roles in endothelial cell function. The present study elucidated the potential VED-associated mechanisms in diabetes, with a specific focus on the complicated mutual effects between *circHMGCS1* and *MIR4521*. The ceRNA network involving *circHMGCS1-MIR4521-ARG1* can become a novel and significant regulatory pathway for preventing and potentially treating diabetes-induced

VED. Our findings suggest that targeted inhibition of circHMGCS1 or the overexpression of *MIR4521* could serve as effective strategies in mitigating diabetes-induced VED. These insights underscore the promising function of ncRNAs in developing therapeutic interventions for diabetes-associated VED.

## Materials and methods

**Key resources table**

| Reagent type (species) or resource | Designation | Source or reference | Identifiers | Additional information |
|---|---|---|---|---|
| Cell line (*Homo sapiens*) | HUVEC | The Shanghai Cell Bank of the Chinese Academy of Sciences | RRID:CVCL_2959 | |
| Cell line (*H. sapiens*) | HEK-293T | The Shanghai Cell Bank of the Chinese Academy of Sciences | RRID:CVCL_0063 | |
| Gene (*H. sapiens*) | circHMGCS1 | circbase | circRNA ID: hsa_circ_0008621 | |
| Gene (*H. sapiens*) | HMGCS1 | GeneBank | Gene ID: 3157 | |
| Gene (*H. sapiens*) | MIR4521 | miRbase | miRNA ID: MIMAT0019058 | |
| Gene (*Mus musculus*) | ARG1 | GeneBank | Gene ID: 383 | |
| Gene (*M. musculus*) | Arg1 | GeneBank | Gene ID: 11846 | |
| Antibody | Anti- Liver Arginase (ARG1) (Rabbit monoclonal) | Abcam | ab133543 | WB (1: 1000) |
| Antibody | Anti- Arg2 (Rabbit monoclonal) | Abcam | ab264066 | WB (1: 1000) |
| Antibody | Anti- Endothelin 1(ET-1) (Mouse monoclonal) | Abcam | ab2786 | WB (1: 1000) |
| Antibody | Anti- VCAM1 (Rabbit monoclonal) | Abcam | ab134047 | WB (1: 1000) |
| Antibody | Anti-ICAM1 (Rabbit monoclonal) | Abcam | ab222736 | WB (1: 1000) |
| Antibody | Anti-AGO2 (Mouse monoclonal) | Proteintech | 67934–1-Ig | WB (1: 2000) |
| Antibody | Anti-beta ACTIN (Mouse monoclonal) | Abcam | ab8226 | WB (1: 2000) |
| Antibody | Anti-GAPDH (Mouse monoclonal) | Abcam | ab8245 | WB (1: 2000) |
| Recombinant DNA reagent | pCDH-CMV-MCS-EF1-Puro (plasmid) | Addgene | RRID:Addgene_73030 | |
| Recombinant DNA reagent | pLKO.1-GFP-shRNA (plasmid) | Addgene | RRID:Addgene_30323 | |
| Recombinant DNA reagent | psPAX2(plasmid) | Addgene | RRID:Addgene_12260 | |
| Recombinant DNA reagent | pMD2G(plasmid) | Addgene | RRID:Addgene_12259 | |
| Recombinant DNA reagent | pAAV-MCS (plasmid) | Ruipute | Cat#: 2212E4 | |
| Recombinant DNA reagent | pAAV-RC9 (plasmid) | Ruipute | Cat#: 23011 | |
| Recombinant DNA reagent | pHelper (plasmid) | Ruipute | Cat#: 230112 | |
| Recombinant DNA reagent | pAAV-MIR4521 sponge-zsGREEn1-shRNA(plasmid) | Ruipute | Cat#: 230113 | |
| Recombinant DNA reagent | pAAV-Arg1-zsGREEn1-shRNA(plasmid) | Ruipute | Cat#: 2212E1 | |

*Continued on next page*

*Continued*

| Reagent type (species) or resource | Designation | Source or reference | Identifiers | Additional information |
|---|---|---|---|---|
| Recombinant DNA reagent | pAAV-ARG1-zsGREEn1-shRNA(plasmid) | Ruipute | Cat#: 2212E2 | |
| Recombinant DNA reagent | pAAV-ZsGreen1 (plasmid) | Takara | Cat#: 6231 | |
| Recombinant DNA reagent | pAAV-ZsGreen1-shRNA (plasmid) | youbio | Cat#: VT8093 | |
| Commercial assay or kit | Fluorescent In Situ Hybridization Kit | RIBOBIO | Cat#: C10910 | |
| Commercial assay or kit | RNA pull down kit | GENESEED | Cat#: P0202 | |
| Commercial assay or kit | RNA Immunoprecipitation Kit | GENESEED | Cat#: P0102 | |
| Software, algorithm | ImageJ | National Institutes of Health | https://imagej.nih.gov/ij/ | |
| Software, algorithm | Prism | GraphPad v8.0 | RRID:SCR_002798 | |

## Cell culture

HUVECs were cultured in high glucose Dulbecco's Modified Eagle Medium (DMEM; Hyclone, USA) supplemented with 10% fetal bovine serum (FBS; Gibco, USA), 100 µg/mL of streptomycin, and 100 U/mL of penicillin. The cells were cultured at 37 °C in a humidified incubator with 95% air and 5% $CO_2$. To simulate the elevated glucose levels observed in diabetes, we cultured HUVECs in a PAHG medium. The medium contained 25 mM glucose and 250 µM saturated free fatty acid (FFA) palmitate (16:C; Sigma, USA) and was incubated for 24 hr. For in vitro experiments, HUVECs were subcultured in six-well plates. The cells were then transfected with *MIR4521* mimics, MIR-inhibitor, or MIR-NC and incubated for 48 hr. Subsequently, the cells were treated with PAHG for another 24 hr. Samples were collected to determine the corresponding indices.

## Animal design

Male C57BL/6 J mice (6–8 weeks old) were obtained from SLAC Laboratory Animal Co., Ltd. (SLAC ANIMAL, China). The mice were housed at a temperature of 22 ± 1 °C with a 12 hr light/dark cycle. To induce diet-induced diabetes, we fed wild type littermates either a standard chow (Control) or a high fat-high sucrose (HFHG) diet, where the diet composition consisted of 60% fat, 20% protein, and 20% carbohydrate (H10060, Hfkbio, China). The dietary regimen was maintained for 14 weeks. Throughout this period, body weight and fasting blood glucose (FBG) levels were measured on a weekly basis. An FBG level of ≥11.1 mmol/L was used as the criterion for a successful diabetic model. To study the role of *ARG1* in diabetes-induced endothelial dysfunction, we established an HFHG-induced diabetic model (DM), and then pAAV9-*Arg1* shRNA (1×10$^{12}$ vg/mL, 100 µL) was injected into the tail vein. Subsequently, we observed changes in relevant indicators. To investigate the regulatory relationship among *Arg1*, *MIR4521*, and *circHMGCS1* in diabetes-induced endothelial dysfunction, we injected *MIR4521* agomir or pAAV9-*circHMGCS1* via the tail vein, following the procedure described in *Figure 5—figure supplement 1A*. The mice were sacrificed after a 12 hr fast, and blood and other tissues were collected and stored at –80 °C until further analysis. All animal experiments were conducted in strict accordance with the guidelines and laws governing the use and care of laboratory animals in China (GB/T 35892–2018 and GB/T 35823–2018) and NIH Guide for the Care and Use of Laboratory Animals. The experimental procedures involving animals were performed at the Animal Experiment Center of Zhejiang Chinese Medical University in Hangzhou, China. The animal protocol was conducted in compliance with the guidelines set forth by the Ethics Committee for Laboratory Animal Care at Zhejiang Chinese Medical University (Approval No. 2022101345).

## Vascular function

After sacrificing the mice, the thoracic aortae and mesenteric arteries were carefully removed and dissected in an oxygenated ice-cold Krebs solution. The changes in the isometric tone of the aortic

rings and second-order mesenteric arteries were recorded using a wire myograph (Danish Myo, DK). The arterial segments were stretched to achieve an optimal baseline tension (3 mN for the aorta) and allowed to equilibrate for 1 hr. Subsequently, the segments were contracted using 60 mmol/L of KCl and rinsed with Krebs solution. Endothelium-dependent relaxation (EDR) was evaluated by assessing the concentration-responses to cumulative additions of acetylcholine (ACh) in noradrenaline (NE, $10^{-3}$ mol/L) precontracted rings.

## Blood pressure measurements

An intelligent non-invasive blood pressure monitor (BP-2010AUL) was used to measure blood pressure in different mouse groups simultaneously. Prior to measurement, the mice were placed in a blood pressure room until reaching a temperature of 25 ± 1 °C. After a 15 min equilibration period, the mice were gently positioned in a mice jacket set at a constant temperature of 40 °C, with their tail inserted into a pulse sensor. Throughout the measurements, a quiet environment was maintained to minimize external disturbances. The instrument automatically recorded the stabilized sensor signal to obtain the systolic blood pressure (SBP).

## Serum lipid assays

After 5 weeks of blood collection from mouse orbits and 12 weeks of drug administration, blood samples were obtained from mouse arteries. The collected blood was centrifuged at 1500 × *g* for 15 min to separate the serum. Commercial kits and an automatic biochemical analyzer (Biobase BK-600) were used to measure triglycerides (TG), total cholesterol (TC), high-density lipoprotein cholesterol (HDL-c), and low-density lipoprotein cholesterol (LDL-c).

Construction of *circHMGCS1* and *ARG1* plasmids and stable transfection using lentivirus *circHMGCS1*-overexpressing lentiviruses (pLV-*circHMGCS1*) were obtained from Hangzhou Ruipu Biotechnology Co., Ltd (Guangzhou, China). The *circHMGCS1* sequence [NM_001098272: 43292575–43297268], the splice site AG/GT and ALU elements were inserted into the pCDH-circRNA-GFP vector (upstream ALU: AAAGTGCTGAGATTACAGGCGTGAGCCACCACCCCCGGCCCACTTTTT GTAAAGGTACGTACTAATGACTTTTTTTTTTATACTTCAG, downstream ALU: GTAAGAAGCAAGGAAA AGAATTAGGCTCGGCACGGTAGCTCACACCTGTAATCCCAGCA). The restriction enzyme sites selected were EcoRI and NotI. Lentiviruses containing an empty vector and expressing GFP were used as the negative control (pLV-NC; *Liang and Wilusz, 2014*). *ARG1*-overexpressing lentiviruses (pLV-*ARG1* OE) were acquired from Genomeditech (China). HUVECs were cultured in six-well plates until reaching 60% confluence and then infected with lentivirus particles in the presence of 5 μg/ml of polybrene at a multiplicity of infection of 30. The cells were cultured for at least 3 days before further experiments were conducted. The overexpression efficiency of *circHMGCS1* or *ARG1* was evaluated using quantitative real-time PCR or western blottings.

## Cloning and production of *circHMGCS1*, *MIR4521* sponge, *circHMGCS1* shRNA and *ARG1* shRNA

The pAAV-*circHMGCS1*-ZsGreen1 vector was constructed by inserting the full-length circHMGCS1 into the pAAV-circRNA-ZsGreen1 vector. *circHMGCS1* shRNA vector was constructed by inserting *circHMGCS1* shRNA (5'-ACAUAGCAACUGAGGGCUUCG-3') into pLKO.1 GFP shRNA plasmid. To achieve AAV-mediated *ARG1* gene silencing, we obtained a shRNA sequence targeting *ARG1* from Sigma-Aldrich Mission RNAi (TRCN0000101796). The oligo sequences 5'GATCCGCCTTTGTTGA TGTCCCTAATCTCGAGATTAGGGACATCAACAAAGGCTTTTTA-3' and 5'-AGCTTAAAAAGCCTTT GTTGATGTCCCTAATCTCGAGATTAGGGACATCAACAAAGGCG-3' were synthesized, annealed, and ligated to the pAAV-ZsGreen-shRNA shuttle vector, resulting in the construction of pAAV-shARG1-ZsGreen1. The *MIR4521* sponge cassette was cloned into the pAAV-ZsGreen1 expression vector to generate pAAV-*MIR4521* sponge-ZsGreen1. Recombinant AAVs (rAAV9s) were produced by transfecting HEK293T cells plated at a density of 1×10⁷ cells/15 cm plate 1 day prior to transfection using polyethyleneimine (Linear PEI, MW 25 kDa, Sigma-Aldrich) as the transfection reagent. For each 15 cm plate, the following reagents were added: 20 μg of pHelper, 10 μg of pAAV-RC9, and 10 μg of pAAV-circHMGCS1-ZsGreen1, pAAV-shARG1-ZsGreen1, or pAAV-*MIR4521* sponge-ZsGreen1. These plasmids were combined with 500 μL of serum-free and antibiotic-free DMEM and 100 μL of PEI reagent (1 mg/mL, pH 5.0). The DNA-PEI reagent was added drop-wise to the cells without changing

the media on the 15 cm plates. After three days, the cells were collected and lysed through repeated cycles of freezing and thawing at −80 °C and 37 °C, respectively. The rAAV9s were purified using the ViraTrap Adenovirus Purification Maxiprep Kit (BW-V1260-02, Biomiga). The virus titer was determined using qPCR and adjusted to $1 \times 10^{12}$ viral genomes (vg)/mL in PBS containing 4% sucrose.

## Transfection of miRNA mimic or inhibitor

*MIR4521* mimics, miRNA negative control (MIR-NC), *MIR4521* inhibitor, and miRNA inhibitor negative control (MIR-NC inhibitor) were obtained from Tingske (China). To investigate the functional role of *MIR4521* in HUVECs, we performed transfections using Hieff Trans in vitro siRNA/miRNA Transfection Reagent with *MIR4521* mimics, MIR-NC, MIR-NC inhibitor, or MIR-NC inhibitor at a concentration of 50 nM, following the manufacturer's protocols. After 48 hr of transfection, proteins and RNA were collected for further analysis. The sequences used in the experiments are listed as follows: miRNA negative control (MIR-NC): 5′-UUCUCCGAACGUGUCACGUTT-3′, 5′-ACGUGACACGUUCGGAGAAT T-3′;*MIR98-5P*-mimics: 5′-UGAGGUAGUAAGUUGUAUUGUU-3′, 5′-CAAUACAACUUACUACCUCA UU-3′; *MIR3143*-mimics: 5′-UAACAUUGUAAAGCGCUUCUUU-3′, 5′-AGAAGCGCUUUACAAUGUUA UU-3′; *MIR181A–2–3* P-mimics: 5′-ACCACUGACCGUUGACUGUACC-3′, 5′-UACAGUCAACGG UCAGUGGUUU-3′; *MIR4521*-mimics: 5′-GAGCACAGGACUUCCUUAGCUU-3′, 5′-GCUAAGGA AGUCCUGUGCUCAG-3′; *MIR4521*-inhibitor: 5′-CUGAGCACAGGACUUCCUUAGC-3′; miRNA inhibitor negative control (MIR-NC inhibitor): 5′-CAGUACUUUUGUGUAGUACAA-3′; *MIR4521* agomir: 5′-GAGCACAGGACUUCCUUAGCUU: 5′-GCUAAGGAAGUCCUGUGCUCAG-3′.

## NO assays

The supernatant fluid from HUVECs treated with the indicated reagents and a tissue homogenate of aortic rings were collected. NO concentrations were determined using the Griess reagent and a total nitric oxide assay kit (Beyotime, China) following the manufacturer's instructions.

## *ENOS* activity assay

The cellular *ENOS* activity was assessed as per previously described protocol. (*Leopold et al., 2007*). Briefly, Cells were exposed to a PBS buffer (400 µL/well) containing 1.5 Ci/ml [3 H] L-arginine for 15 min to initiate *ENOS* activity measurement. The reaction was stopped using 1 N ice-cold TCA (500 µL/well). After freeze-thawing in liquid nitrogen, cells were scraped, treated with water-saturated ether three times, and neutralized with 1.5 ml of 25 mM pH 8.0 HEPES. Dowex AG50WX8 columns (Tris form) were used for elution with a 1 ml solution of 40 mM pH 5.5 HEPES supplemented with 2 mM EDTA and 2 mM EGTA. The eluate was collected for [3 H] L-citrulline quantification using liquid scintillation spectroscopy. Additionally, the assessment of ENOS activity in cell lysates and tissues was performed through the previously outlined immunoprecipitation approach (*Du et al., 2001*). Briefly, the samples were divided into two tubes: one for protein blotting and the other for *ENOS* activity measurement. The *ENOS* immunocomplexes, linked to protein A-Sepharose beads, were reconstituted in assay buffer, and *ENOS* activity was gauged by tracking the transformation of [3 H] L-arginine into [3 H] L-citrulline. Equal enzyme quantities in each incubation were validated via western blotting.

## Detection of ROS for tissue and cell

To assess the level of ROS in tissues, we used a red fluorescent reactive oxygen probe called DHE (dihydroethidium; *Karim et al., 2022*; *Su et al., 2018*). In brief, 10 µL of the DHE probe was combined with 190 µL of tissue homogenate supernatant, thoroughly mixed, and subsequently incubated in darkness at 37 °C for 30 min. Following incubation, the samples were promptly imaged under a fluorescent microscope.

For ROS detection in cells, the treated cells were washed once by PBS, then 20 µM DHE was added, and incubated at 37 °C for 30 min away from light, then washed three times by PBS and then colorless DMEM medium was added, followed by fluorescence microscopy for observation.

## CircRNA sequencing

Total RNA was extracted from HUVECs, both with and without PAHG stimulation, utilizing Trizol (Invitrogen, Carlsbad, USA) following the manufacturer's instructions. The extracted total RNA was then assessed for quality and quantity using a denaturing agarose gel, Nano Drop, and Agilent 2100

bioanalyzer (Thermo Fisher Scientific, USA). For the construction of RNA-seq libraries to obtain sequence information of linear transcripts, 3 µg of total RNA samples underwent treatment with the epicenter Ribo-ZeroTM Kit (Illumina, San Diego, USA) to remove rRNA. Linear RNA was subsequently eliminated using RNase R (Epicentre Technologies, USA). The resulting cleaved RNA fragments were reverse-transcribed to generate cDNA, which served as the template for the synthesis of U-labeled second-stranded DNAs. This synthesis step involved the use of *E. coli* DNA polymerase I, RNase H, and dUTP, facilitated by the PrimeScript RT regent Kit (TaKaRa, Japan). Purification of the synthesized second-stranded DNAs was accomplished using AMPureXP beads. All subsequent steps were conducted in accordance with the manufacturer's protocols. To assess the quality of the libraries, an Agilent 2100 Bioanalyzer was employed, and the paired-end sequencing was performed using an Illumina Hiseq 4000 (LC Bio, China), adhering to the recommended protocol provided by the vendor.

## Treatment of circRNA raw sequencing data

The sequencing data was filtered with Cutadapt by removing reads containing adaptor contamination, poly-N or low quality and undetermined bases, and then sequence quality was verified using FastQC (http://www.bioinformatics.babraham.ac.uk/projects/fastqc/) (*Kechin et al., 2017*). Bowtie2 and Hisat2 to map reads to the genome of species (*Kim et al., 2015*; *Langmead and Salzberg, 2012*). CIRCExplorer2 and CIRI were used to de novo assemble the mapped reads to circular RNAs (*Kim and Salzberg, 2011*; *Zhang et al., 2016*). Then, back splicing reads were identified in unmapped reads by tophat-fusion. All samples were generated unique circular RNAs. The expression level was calculated according to the junction reads per billion mapped reads (RPB). Differentially expressed circRNAs analysis was performed with log2 (fold change)>1 or log2 (fold change) <-1 and with statistical significance (p-value <0.05) by R package–edgeR (*Robinson et al., 2010*).

## microRNA microarray assay

Total RNA was extracted from HUVECs treated with either PAHG or PBS using TRIzol reagent (Invitrogen, USA). The µParaflo MicroRNA microarray Assay by LC Sciences (LC Sciences, USA) using miRBase version 22.0 was performed. After 3'-extension with a poly(A) tail and ligating an oligonucleotide tag, hybridization was done overnight on a µParaflo microfluidic chip. Cy3 dye facilitated dye staining after RNA hybridization. Fluorescence images were acquired using the GenePix 400B laser scanner (Molecular Devices, USA). Data were analyzed by background subtraction and signal normalization using a LOWESS filter. miRNA expression with a two-tailed p-value of <0.05 was considered significant. Intensities below 500 were considered false positives.

## Network analysis of competing endogenous RNA

The miRNA targets of *circHMGCS1* were computationally predicted using three bioinformatic tools: miRanda (http://www.microrna.org/microrna/home.do), whereas the potential target genes of *MIR4521* were envisaged using four distinct databases: TargetScan (http://www.targetscan.org/), GenCard (https://www.genecards.org/), miRWalk (http://mirwalk.umm.uni-heidelberg.de/), and mirDIP (http://ophid.utoronto.ca/mirDIP/).

## Isolation of RNA and miRNA for quantitative real time-PCR (qRT-PCR) analysis

RNA was extracted utilizing a total RNA extraction kit (Yeason, 19221ES50, China) and a miRNA extraction kit (Yeason, 19331ES50, China) according to the corresponding manufacturer's protocol. The RNA concentration was determined at 260 nm using a spectrophotometer. Reverse transcription was conducted using a reverse transcription kit (TAKARA, Japan) with random primers according to the manufacturer's protocol to synthesize cDNA and circRNA. Quantitative real-time PCRs (qRT-PCRs) were conducted on the CFX96 Touch Real-Time PCR Detection System (Bio-Rad, USA) using the standard protocol provided by the SYBR Green PCR Kit (Yeason, China). For miRNA expression detection, reverse transcription was executed, and microRNAs were detected using stem-loop primers procured from Tsingke (China). The relative gene expression was calculated using the $2^{-\Delta\Delta Ct}$ method. All qRT-PCR assays were repeated thrice. The primers used for the qRT-PCR assays are listed as follows: *MIR98-5P*: 5'-CGCGCGTGAGGTAGTAAGTTGT-3', 5'-AGTGCAGGGTCCGAGGTATT-3'; *MIR3143*: 5'-GCGA TAACATTGTAAAGCGCTT-3', 5'-AGTGCAGGGTCCGAGGTATT-3'; *MIR181A-2–3* P: 5'-GCGACCAC

TGACCGTTGAC-3', 5'-AGTGCAGGGTCCGAGGTATT-3'; *MIR4521*: 5'-CGGCTAAGGAAGTCCT GTGC-3', 5'-CGCAGGGTCCGAGGTATTC-3'; *U6*: 5'-ATTGGAACGATACAGAGAAGATT-3', 5'-GGAA CGCTTCACGAATTTG-3'; *ET-1*: 5'-GCCTGCCTTTTCTCCCCGTTAAA-3', 5'-CAAGCCACAAACAGCA GAGA-3'; ICAM1: 5'-TGACCGTGAATGTGCTCTCC-3', 5'-TCCCTTTTTGGGCCTGTTGT-3'; *ARG1*: 5'-TTCTCAAAGGGACAGCCACG-3', 5'-CGCTTGCTTTTCCCACAGAC-3'; *ICAM1*: 5'-ATGCCCAGACAT CTGTGTCC-3', 5'-GGGGTCTCTATGCCCAACAA-3'; *VCAM1*: 5'-AAATCGAGACCACCCCAGAA-3', 5'-AGGAAAAGAGCCTGTGGTGC-3'; *β-ACTIN*: 5'-CTCACCATGGATGATGATATCGC-3', 5'-CACA TAGGAATCCTTCTGACCCA-3'; *HMGCS1*: 5'-TCGTGGGACACATATGCAAC-3', 5'-TGGGCATGGATC TTTTTGCAG-3'; *hsa_circ_0000992*: 5'-TCACACGGGAGAGTGTTACC-3', 5'-GTCCATCGAGAAAAGC TGATGC-3'; *hsa_circ_0004036*: 5'-GGCTCACAAGCAGCCTTTAC-3', 5'-GGTGATACAGGAGCGG GTAG-3'; *hsa_circ_0004889*: 5'-GTCATATATGGGAAAGAGGGCTT-3', 5'-GGACAAGCCTTCATGG GCTC-3'; *hsa_circ_0006719*: 5'-CCTCCCTCGGTGTTGCCTTC-3', 5'-TCCAGGCCAGGTAGACAGAA C-3'; *hsa_circ_0008621*: 5'-CAAACATAGCAACTGAGGGCTTC-3', 5'-TAATGCACTGAGGTAGCACT G-3'; *GAPDH*: 5'-GAAAGCCTGCCGGTGACTAA-3', 5'-GCATCACCCGGAGGAGAAAT-3'.

## circRNA validation by PCR

DNA and gDNA templates were PCR amplified using BioRad Mastercyclers following the manufacturer's protocol. Subsequently, the PCR products were visualized using a 1% agarose gel stained with GelRed. To validate the PCR results, we purified the products using the EZ-10 Column DNA Purification Kit (Sangon Biotech, China), and direct PCR product Sanger sequencing was performed. *circHMGCS1*-full length: 5'-AGCAACTGAGGGCTTCGT-3',5'-ATGTTTGAATGCACAAGTCCTAC-3';*circHMGCS1*-divergent primers: 5'-TTATGGTTCTGGTTTGGCTGC-3', 5'-TTCAGCGAAGACATCT GGTGC-3'; *circHMGCS1*-convergent primers: 5'-ATAGCAACTGAGGGCTTCGTG-3', 5'-GCGGTCTA ATGCACTGAGGT-3'.

## Cell nuclear/cytoplasmic fractionation

RNA extracts from HUVECs were subjected to nuclear/cytoplasmic isolation using the NE-PER Nuclear and Cytoplasmic Extraction Reagent (Thermo Fisher Scientific, USA) following the manufacturer's protocol.

## RNase R treatment

For RNase R treatment, approximately 2 μg of total RNAs from HUVECs were incubated with or without 20 U of RNase R (Geneseed, China) at 37 °C for 30 min. The resulting RNAs were then purified using an RNA Purification Kit (QIAGEN, USA).

## Actinomycin D assay

HUVECs were seeded equally in 24-well plates at a density of $5 \times 10^4$ cells per well. Subsequently, the cells were exposed to actinomycin D (2 μg/mL; Abcam, USA) for different time intervals: 0, 4, 8, 12, and 24 hr. After the respective exposure times, the cells were collected for RNA extraction. The relative RNA levels of circHMGCS1 and HMGCS1 mRNA were analyzed using qRT-PCR and normalized to the values obtained from the control group (0 hr).

## RNA immunoprecipitation

RIP assay was conducted using the PureBindingRNA Immunoprecipitation Kit (Geneseed, China) as per the manufacturer's guidelines. HUVECs were lysed in complete RNA immunoprecipitation lysis buffer containing 1% proteinase inhibitor and 1% RNase inhibitor after being infected with circNC or *circHMGCS1*. The cell extract was then incubated with magnetic beads conjugated with anti-Argonaute 2 (*AGO2*) or anti-IgG antibody (Abcam, USA) overnight at 4 °C. The beads were subsequently washed, and the proteins were removed using columns. Eventually, the isolated RNA was extracted using TRIzol Reagent, and the purified RNA was utilized for subsequent qRT-PCR analysis.

## Pull-down assay

A pull-down assay was conducted using the PureBinding RNA-Protein pull-down Kit (Geneseed, P0201, China) following the manufacturer's protocol. The pull-down assay with biotinylated *MIR4521* or *circHMGCS1* was performed as previously described (*Wang et al., 2020*). Briefly, HUVECs were

transfected with biotinylated RNA or mutants (Tingke, 50 nmol/L) using the Hieff Trans in vitro siRNA/miRNA Transfection Reagent (Yeason, China). Following a 48 hr transfection period, the cells were collected, PBS-washed, and incubated for 10 min in a capture buffer on ice. A 10% fraction of the cell lysates was retained as input. The remaining lysates were subsequently exposed to streptavidin magnetic beads for 30 min at 4 °C. After treatment with wash buffer and the RNeasy Mini Kit (QIAGEN), the captured RNAs were extracted for subsequent qRT-PCR analysis,while the associated proteins were processed for western blot analysis.

The sequences are listed as follows: Biotin-*MIR4521*-MUT: 5'-GCAUUCCUUCAGGAGUGCUCAG-3', Biotin-*MIR4521*-WT: 5'-GCUAAGGAAGUCCUGUGCUCAG-3'. Biotin-*circHMGCS1*-WT: 5'-ATGTGTCCCACGAAGCCCTCAGTTGCTATGTTTGAA-3', Biotin-*circHMGCS1*-MUT: 5'- GACGTCGTGTGCGTCGGTGCTAAGCTTCACAGATAC-3'.

## Western blot analysis

Western blot was performed following previously described procedures (*Xu et al., 2022*). Protein extraction from cells or tissues was carried out using protein lysis buffer containing protease and phosphatase inhibitor cocktail (Beyotime, China). The concentrations of the protein lysates were determined using the Beyotime BCA Protein Assay Kit (Beyotime, China) according to the manufacturer's instructions. Equal quantities of proteins (20 µg/lane) were loaded onto 8% or 10% SDS-PAGE gels and subsequently transferred to PVDF membranes (Millipore, USA). The membranes were then incubated with primary antibodies overnight at 4 °C. The primary antibodies used were as follows: anti-*ARG1* (diluted 1:1000; Abcam, ab133543), anti-*ARG2* (diluted 1:1000; Abcam, ab264066), anti-*ET-1* (diluted 1:1000; Abcam, ab2786), anti-*VCAM1* (diluted 1:1000; Abcam, ab134047), anti-*ICAM1* (diluted 1:1000; Abcam, ab222736), anti-AGO2(diluted 1:2000; proteintech, 67934–1-Ig), anti-*β-ACTIN* (diluted 1:2000; Abcam, ab8226), and anti-*GAPDH* (diluted 1:2000; Abcam, ab8245). After incubation with the corresponding secondary antibodies for 2 hr at room temperature, the membranes were washed. Antibody binding was detected using an ECL detection reagent (Millipore, USA). Digital images were captured using a Gel DocTM XR +System with ImageLab software (Bio-Rad, USA). The quantification of the labeled bands was performed using ImageJ 1.55.

## RNA fluorescent in situ hybridization (RNA-FISH)

The RNA-FISH assay was conducted in HUVECs. Cy3-labeled *circHMGCS1* probe and FAM-labeled *MIR4521* probe were custom-designed and synthesized by RiboBio (China). The signals emanating from the probes were detected using the Ribo FISH Kit (RiboBio) in accordance with the manufacturer's protocol. Briefly, frozen sections were fixed with 4% paraformaldehyde for 10 min, followed by PBS washing. The fixed sections were treated with PBS containing 0.5% triton X-100 at 4 °C for 5 min, followed by incubation in pre-dehydration buffer at 37 °C for 30 min. Subsequently, probes targeting *circHMGCS1* and *MIR4521* were applied to the sections. Hybridization was performed in a humid chamber at 37 °C overnight. Post-hybridization washing was initially conducted with 4×saline sodium citrate containing 0.1% tween-20 at 42 °C, followed by further washing with 2×saline sodium citrate at 42 °C to eliminate non-specific and repetitive RNA hybridization. The slides were counterstained with DAPI (beyotime, China) and examined using an Olympus FV1200 (Olympus, Japan) confocal microscopy system. The mean fluorescent intensity (MFI) was determined using Image J software and subjected to statistical analysis. The probe sequences are detailed as follows: Cy3-*circHMGCS1*: 5'-UCCCACGAAGCCCUCAGUUGCUAUG-3', FAM-*MIR4521*: 5'-UUUGACUCGUGUCCUGAAGGAAUCG-3'.

## Luciferase reporter assay

The complete sequence of *circHMGCS1* and a mutant *circHMGCS1* sequence lacking the miRNA binding site were chemically synthesized. These sequences were inserted into the Nhel and SalI sites of the pmirGLO vector to generate expression vectors denoted as *circHMGCS1*-WT and *circHMGCS1*-MUT. Transfection was carried out using lipofectamine 2000 (Invitrogen, USA) with 50 nM *MIR4521* mimics, MIR-NC, *MIR4521* inhibitor, and MIR-NC inhibitor. HEK293T cells were seeded in 96-well plates and cultured until reaching 50%–70% confluence prior to transfection. After 48 hr of transfection, cells were harvested, and luciferase activity was quantified utilizing the Dual Luciferase Reporter Gene Assay Kit (Yeason, China). The pmirGLO vector expressing Renilla luciferase

was utilized as an internal control for transfection, while the empty pmirGLO vector served as the negative control. Firefly luciferase activity of the pmirGLO vector was normalized with Renilla luciferase activity for comparison. Luciferase reporter assay was also used to investigate the regulation of *MIR4521* on the expression of its target genes. For ARG1 and *MIR4521*, either wild-type or mutant *ARG1* 3'UTR fragments (433 bp) were inserted into NheI/SalI restriction sites of pmirGLO. One μg plasmids of *ARG1* 3'UTR-WT and *ARG1* 3'UTR-MUT, 50 nM *MIR4521* mimics, MIR-NC, *MIR4521* inhibitor and MIR-NC inhibitor were transfected. Luciferase activity was evaluated 48 hr post-transfection using the Dual Luciferase Reporter Gene Assay Kit (Yeason).

## Histological and morphometric analysis

Aortic tissues were prepared by fixing them in 4% paraformaldehyde in PBS and subsequently embedding them in paraffin. The resulting paraffin-embedded tissues were then sectioned into 4-μm-thick slices. Hematoxylin-eosin (H&E) staining was performed on these sections to visualize the vascular tissue structures and assess the degree of thickness in the thoracic aorta. The average thickness of five randomly selected regions of the thoracic aorta was measured using OlyVIA software (Olympus, Japan).

## Statistical analysis

Statistical analyses were conducted using Prism software (GraphPad 8.0). Data are presented as mean ± standard deviation (SD) unless otherwise stated. Prior to statistical analysis, the normality of data distribution was evaluated using the Shapiro-Wilk test. For comparisons between the two groups, an unpaired two-tailed Student's t-test was used. When comparing more than the two groups, either a one-way or two-way analysis of variance (ANOVA) was conducted, followed by the Bonferroni multiple comparison post hoc test. Group differences were considered statistically significant at a p-value of $<0.05$. The level of significance was indicated by asterisks, with *, **, and *** denoting p-values lower than 0.05, 0.01, and 0.001, respectively.

## Acknowledgements

The authors thank core facilities of medicine in Zhejiang university for technical support. Funding: This work was supported by the National Natural Science Foundation of China (32172192) and Zhejiang Provincial Key R&D Program of China (2021C02018).

# Additional information

## Funding

| Funder | Grant reference number | Author |
|---|---|---|
| The National Natural Science Foundation of China | 32172192 | Wei Chen |
| The National Natural Science Foundation of China | 2021C02018 | Wei Chen |

The funders had no role in study design, data collection and interpretation, or the decision to submit the work for publication.

## Author contributions

Ming Zhang, Conceptualization, Data curation, Formal analysis, Validation, Investigation, Methodology, Writing – original draft, Writing – review and editing; Guangyi Du, Data curation, Investigation, Methodology; Lianghua Xie, Yang Xu, Investigation, Methodology; Wei Chen, Conceptualization, Resources, Supervision, Funding acquisition, Methodology, Writing – original draft, Writing – review and editing

## Author ORCIDs

Lianghua Xie https://orcid.org/0009-0004-1669-4593
Wei Chen https://orcid.org/0000-0002-2373-2437

## Ethics

All animal experiments were conducted in strict accordance with the guidelines and laws governing the use and care of laboratory animals in China (GB/T 35892-2018 and GB/T 35823-2018) and NIH Guide for the Care and Use of Laboratory Animals. The experimental procedures involving animals were performed at the Animal Experiment Center of Zhejiang Chinese Medical University in Hangzhou, China. The animal protocol was conducted in compliance with the guidelines set forth by the Ethics Committee for Laboratory Animal Care at Zhejiang Chinese Medical University (Approval No. 2022101345).

Reviewer #1 (Public Review): https://doi.org/10.7554/eLife.97267.3.sa1
Reviewer #2 (Public Review): https://doi.org/10.7554/eLife.97267.3.sa2
Author response https://doi.org/10.7554/eLife.97267.3.sa3

# Additional files

## Supplementary files

• MDAR checklist

## Data availability

Sequencing data have been deposited in GEO under accession codes GSE237295 and GSE237597, All data generated or analysed during this study are included in the manuscript and supporting files. Source data files for gels and blots are provided in the manuscript and supporting files.

The following datasets were generated:

| Author(s) | Year | Dataset title | Dataset URL | Database and Identifier |
|---|---|---|---|---|
| Chen W, Zhang M | 2024 | The Circular RNA HMGCS1 Sponges miR-4521 to Aggravate Diabetes-Induced Vascular Endothelial Dysfunction Through Upregulating ARG1 | https://www.ncbi.nlm.nih.gov/geo/query/acc.cgi?acc=GSE237295 | NCBI Gene Expression Omnibus, GSE237295 |
| Chen W, Zhang M | 2024 | The Human Circular RNA CircHMGCS1 Sponges miR-4521 to Aggravates Diabetes-Induced Vascular Endothelial Dysfunction Through Upregulating ARG1 | https://www.ncbi.nlm.nih.gov/geo/query/acc.cgi?acc=GSE237597 | NCBI Gene Expression Omnibus, GSE237597 |

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
