## [Editor Report · eLife assessment]

This study presents **important** findings linking circHMGCS1 and miR-4521 in diabetes-induced vascular endothelial dysfunction. Overall, the evidence supporting the claims of the authors is **convincing**. The work will be of interest to biomedical scientists working with cardiovascular and/or RNA biology, particularly those studying diabetes.

---

## [Referee Report · Reviewer #1 (Public Review)]

This study presents a valuable finding on the expression levels of circHMGCS1 regulating arginase-1 by sponging miR-4521observed in diabetes-induced vascular endothelial dysfunction, leading to decrease in vascular nitric oxide secretion and inhibition of endothelial nitric oxide synthase activity. Further, increase in the expression of adhesion molecules and generation of cellular reactive oxygen species reduced vasodilation and accelerated the impairment of vascular endothelial function.

Modulating circHMGCS1/miR-4521/ARG1 axis could serve as a potential strategy to prevent diabetes-associated cardiovascular diseases.

Comments on revised version:

The authors answered all questions satisfactorily.

---

## [Referee Report · Reviewer #2 (Public Review)]

Summary:

The authors observed an aggravated vascular endothelial dysfunction upon overexpressing circHMGCS1 and inhibiting miR-4521. This study discovered that circHMGCS1 promotes arginase 1 expression by sponging miR-4521, which accelerated the impairment of vascular endothelial function.

Strengths:

The study is systematic and establishes the regulatory role of the circHMGCS1-miR-4521 axis in diabetes-induced cardiovascular diseases.

Weaknesses:

(1) The authors show direct evidence of interaction between circHMGCS1 and miR-4521 by pulldown assay. However, the changes in miRNA expression opposite to the levels of target circRNA could be through Target RNA-Directed MicroRNA Degradation. Since the miRNA level is downregulated, the downstream target gene is expected to be upregulated even in the absence of circRNA.

---

## [Author Response]

The following is the authors’ response to the original reviews.

**Public Reviews:**

**Reviewer #1 (Public Review):**
Summary:HMGCS1, 3-hydroxy-3-methylglutaryl-CoA synthase1 is predicted to be involved in Acetyl-CoA metabolic process and mevalonate-cholesterol pathway. To induce diet-induced diabetes, they fed wild-type littermates either a standard chow (Control) or a high fat-high sucrose (HFHG) diet, where the diet composition consisted of 60% fat, 20% protein, and 20% carbohydrate (H10060, Hfkbio, China). The dietary regimen was maintained for 14 weeks. Throughout this period, body weight and fasting blood glucose (FBG) levels were measured on a weekly basis. Although the authors induced diabetes with a diet also rich in fat, the cholesterol concentration or metabolism was not investigated. After the treatment, were the animals with endothelial dysfunction? How was the blood pressure of the animals?

Thank you for your comments and kind suggestions. We have conducted a study on the impact of HFHG diet on the serum levels of total cholesterol(T-CHO) in mice over a 14-week period. Our findings indicated that the HFHG diet significantly elevated T-CHO levels in the serum of mice (Supplementary Figure 5E). Additionally, HFHG diet was associated with an increased in blood pressure (Figure 5F) and it exacerbated the progression of endothelial dysfunction in mice (Figure 5H-L).

Strengths:To explore the potential role of circHMGCS1 in regulating endothelial cell function, the authors cloned exons 2-7 of HMGCS1 into lentiviral vectors for ectopic overexpression of circHMGCS1 (Figure S2). The authors could use this experiment as a concept proof and investigate the glucose concentration in the cell culture medium. Is the pLV-circ HMGCS1 transduction in HUVEC increasing the glucose release? (Line 163)

In the manuscript, we utilized a DMEM culture medium containing 4500 mg/L glucose. Given that the HUVEC cell culture is glucose-dependent for its metabolic processes, it was challenging to precisely evaluate the relationship between pLV-circHMGCS1 transduction and the glucose concentration in the medium.

Weaknesses:(1) Pg 20. The cells were transfected with miR-4521 mimics, miR-inhibitor, or miR-NC and incubated for 24 hours. Subsequently, the cells were treated with PAHG for another 24 hours. Were the cells transfected with lipofectanine? The protocol or the lipofectamine kit used should be described. The lipofectamine protocol suggests using an incubation time of 72 hours. Why did the authors incubate for only 24 hours? If the authors did the mimic and inhibitor curves, these should be added to the supplementary figures. Please, describe the miRNA mimic and antagomir concentration used in cell culture.

For detailed transfection methods of miRNA mimic and its inhibitor, please refer to “Transfection of miRNA mimic or inhibitor” (Line 587) in the revised Experimental Section. We employed the Hieff TranssiRNA/miRNA in vitro transfection reagent (yeason, China, 40806ES03), with a transfection duration of 48h. The miR-4521 content in HUVEC post-transfection was quantified using qRT-PCR. The transfection of the miR-4521 mimic for 48h notably enhanced its expression in HUVEC (Supplementary Figure 3B), whereas the transfection of the miR-4521 inhibitor for the same duration significantly suppressed its expression (Supplementary Figure 3C). The concentration used for both miRNA mimic and inhibitor transfection was 50 nM. In the revised manuscript, we have corrected the transfection time and clarified that we did not utilize miRNA antagomirs in our experiments.

(2) Pg 20, line 507. What was the miR-4521 agomiR used to treatment of the animals?

miRNA agomir serves as a valuable experimental tool for elucidating miRNA function, used to simulate the overexpression of a specific miRNA. miRNA agomir is a chemically modified RNA molecule identical in sequence to the target miRNA, engineered for enhanced stability and transfection efficacy. Utilizing miRNA agomir enables the overexpression of the target miRNA, facilitating the investigation of miRNA functions and mechanism in vivo. In our study, we have employed miRNA mimic for cellular studies and miRNA agomir in vivo applications to achieve high expression of miRNA (Fu et al, 2019).

(3) Figure 1B. The results are showing the RT-qPCR for only 5 circRNA, however, the results show 48 circRNAs were upregulated, and 18 were downregulated (Figure S1D). Why were the other cicRNAs not confirmed? The circRNAs upregulated with high expression are not necessarily with the best differential expression comparing control vs. PAHG groups. Furthermore, Figure 1A and S1D show circRNAs downregulated also with high expression. Why were these circRNAs not confirmed?

Our study aims to the identification of potential biomarkers for endothelial dysfunction in type 2 diabetes, To the end, we focused on circRNAs that exhibited significant upregulation following PAHG treatment. In our sequencing data, the p-values for these top upregulated circRNAs were notably below the threshold of 0.001, prompting their selection for further validation. We employed qRT-PCR to ascertain the consistency of their expression levels with the RNA-sequencing findings. Among these, circHMGCS1 was identified as a promising candidate with regulatory potential in endothelial dysfunction. Additionally, circRNAs that were significantly downregulated will be the subject of our ongoing research endeavors.

(4) Figure 1B shows the relative circRNAs expression. Were host genes expressed in the same direction?

circRNAs are generated from specific exons or introns of their host genes, either individually or in combination, and the main function of circRNA depends on its non-coding RNA characteristics. The expression levels of circRNAs is not necessarily correlated with those of their host genes, and similarly, the function of circRNAs do not inherently relate to the functions of the host genes (Kristensen et al, 2019; Liu & Chen, 2022). Consequently, the data presented in Figure 1B were primarily aimed at validating the accuracy of circRNA-seq. Although we did not conduct host gene expression analysis for the identified circRNAs, our subsequent results indicated that the overexpression of circHMGCS1 did not influence the expression levels of HMGCS1 (Figure 2A).

(5) Line 128. The circRNA RT-qPCR methodology was not described. The methodology should be described in detail in the Methods Session.

The only difference between the circRNA RT-qPCR method and other gene detection is that random primers need to be used for reverse transcription during the reverse transcription process. Unlike linear RNAs that possess a 3' polyA tail, which allows for the use of oligo(dT) primers, circRNAs require random primers to initiate the reverse transcription process. Beyond this distinction, the other processes are no different from the common qRT-PCR process. We have revised the Isolation of RNA and miRNA for quantitative Real Time-PCR (qRT-PCR) analysis method in the revised version (Line 695).

(6) Line 699. The relative gene expression was calculated using the 2-ΔΔCt method. This is not correct, the expression for miRNA and gene expression are represented in percentage of control.

We initially employed the 2^-ΔΔCt method to ascertain the relative gene expression levels. Subsequently, we scaled all values by a factor of 100 to amplify the visual representation of the observed variations, thereby enhancing the visualization of the data.

(7) Line 630. Detection of ROS for tissue and cells. The methodology for tissue was described, but not for cells.

We have added the detailed description of the cellular ROS detection methods in the revised manuscript as follows:

For ROS detection in cells, the treated cells were washed once by PBS, then 20 μM DHE was added, and incubated at 37°C for 30 min away from light, then washed three times by PBS and then colorless DMEM medium was added, followed by fluorescence microscopy for observation (Line 640-643).

(8) Line 796. RNA Fluorescent In Situ Hybridization (RNA-FISH). Figure 1F shows that the RNA-Fluorescence in situ hybridization (RNA-FISH) confirmed the robust expression of cytoplasmic circHMGCS1 in HUVECs (Figure 1F). However, in the methods, lines 804 and 805 described the probes targeting circMAP3K5 and miR-4521 were applied to the sections. Hybridization was performed in a humid chamber at 37C overnight. Is it correct?

We have made a correction in the revised manuscript. The accreted description is "the probes targeting circHMGCS1 and miR-4521 were applied to the sections"(Line816).

(9) Line 14. Fig 1-H. The authors discuss qRT-PCR demonstrated that circHMGCS1 displayed a stable half-life exceeding 24 h, whereas the linear transcript HMGCS1 mRNA had a half-life less than 8 h (Figure 1H). Several of the antibodies may contain trace amounts of RNases that could degrade target RNA and could result in loss of RNA hybridization signal or gene expression. Thus, all of the solutions should contain RNase inhibitors. The HMGCS1 mRNA expression could be degraded over the incubation time (0-24hs) leading to incorrect results. Moreover, in the methods is not mentioned if the RNAse inhibitor was used. Please, could the authors discuss and provide information?

This experiment was performed in cell culture as described in our Experimental Methods (Line 753), where we added actinomycin D directly into the cell culture well plates, and the cells remained in a healthy state during this treatment. We did not directly extract mRNA from cells for this experiment. Additionally, all solutions utilized throughout the whole experiment were prepared using Rnase-free water, ensuring that the integrity of the mRNA.

(10) Further experiments demonstrated that the overexpression of circHMGCS1 stimulated the expression of adhesion molecules (VCAM1, ICAM1, and ET-1) (Figures 2B and 2C), suggesting that circHMGCS1 is involved in VED. How were these genes expressed in the RNA-seq?

In the manuscript, we only focused exclusively on circRNA and miRNA sequencing, and not perform mRNA sequencing, Consequently, we employed qRT-PCR and Western blot to assess the expression alterations of ET-1, ICAM1, and VCAM1 at gene and protein level. The findings revealed that the overexpression of circHMGCS1 significantly upregulated the expression of adhesion molecules (VCAM1, ICAM1, and ET-1).

(11) Line 256. By contrast, the combined treatment of circHMGCS1 and miR-4521 agomir did not significantly affect the body weight and blood glucose levels. OGTT and ITT experiments demonstrated that miR-4521 agomir considerably enhanced glucose tolerance and insulin resistance in diabetic mice (Figures 5C, 5D, and Figures S5B and S5C). Why did the miR-4521 agomir treatment considerably enhance glucose tolerance and insulin resistance in diabetic mice, but not the blood glucose levels?

Our results showed that miR-4521 agomir could effectively suppress the increase of body weight and blood glucose in mice (Figure 5A-B).

(12) In the experiments related to pull-down, the authors performed Biotin-coupled miR-4521 or its mutant probe, which was employed for circHMGCS1 pull-down. This result only confirms the Luciferase experiments shown in Figure 4A. The experiment that the authors need to perform is pull-down using a biotin-labeled antisense oligo (ASO) targeting the circHMGCS1 backsplice junction sequence followed by pulldown with streptavidin-conjugated magnetic beads to capture the associated miRNAs and RNA binding proteins (RBPs). Also, the ASO pulldown assay can be coupled to miRNA RT-qPCR and western blotting analysis to confirm the association of miRNAs and RBPs predicted to interact with the target circRNA.

This point is correct. As suggested, we utilized a biotin-labeled circHMGCS1 probe for pull down experiments. Because circRNA-miRNA interactions are mainly mediated by the RNA-induced silencing complex, which includes Argonaute 2 (AGO2), we examined the levels of miR-4521 and AGO2 in the capture meterial. Our results demonstrated that circHMGCS1 significantly captured miR-4521 in the cells, with a concomitant acquisition of AGO2. These findings have been integrated into the revised manuscript (Supplementary Figures 4D and 4E).

(13) In Figure 5, the authors showed that the results suggest that miR-4521 can inhibit the occurrence of diabetes, whereas circHMGCS1 specifically dampens the function of miR-4521, weakening its protective effect against diabetes. In this context, what are the endogenous target genes for the miR-4521 that could be regulating diabetes?

In this study, we focused on the role of miR-4521 in endothelial function. Our animal experiments involving ARG1 knockdown revealed that the reduction of ARG1 expression resulted in the inability of miR-4521 to modulate the progression of type 2 diabetes. Consequently, ARG1 is likely an endogenous target gene of miR-4521, potentially implicated in the regulation of diabetes.

(14) In the western blot of Figure 5, the β-actin band appears to be different from the genes analyzed. Was the same membrane used for the four proteins? The Ponceau S membrane should be provided.

As described in our experimental methodology (Western blot analysis), we have utilized PVDF membranes for our Western blot experiments. β-actin, recognized for its high expression and specificity as a housekeeping gene, yields distinct bands with minimal background noise. This property can lead to the migration β-actin from the spot wells to both sides during electrophoresis. So much so that it is not aligned with the lane shown by the target gene. And the other 3 genes can see the phenomenon of obvious lane because their expression is not as high as β-actin. We replaced β-actin with a similar background in the revised manuscript (Figure 5L).

(15) Why did the authors use AAV9, since the AAV9 has a tropism for the liver, heart, skeletal muscle, and not to endothelial vessels?

AAV9 has garnered significant interest as a gene delivery vector due to its extensive tissue penetration, minimal immunogenicity, and stable gene expression profile. Its application in cardiovascular disease research and therapy has been widely reported (Barbon et al, 2023; Yao et al, 2018; Zincarelli et al, 2008). Meanwhile, we employed AAV9 for gene delivery via the tail vein injection in mice, and as shown in Figure 5J and Figure 7Q, we observed GFP signals carried by AAV9 in the thoracic aorta of mice. These findings suggest that AAV9 possesses the capability to infect endothelial cells effectively.

**Reviewer #2 (Public Review):**
Summary:The authors observed an aggravated vascular endothelial dysfunction upon overexpressing circHMGCS1 and inhibiting miR-4521. This study discovered that circHMGCS1 promotes arginase 1 expression by sponging miR-4521, which accelerated the impairment of vascular endothelial function.Strengths:The study is systematic and establishes the regulatory role of the circHMGCS1-miR-4521 axis in diabetes-induced cardiovascular diseases.Weaknesses:(1) The authors selected the miR-4521 as the target based on their reduced expression upon circHMGCS1 overexpression. Since the miRNA level is downregulated, the downstream target gene is expected to be upregulated even in the absence of circRNA. The changes in miRNA expression opposite to the levels of target circRNA could be through Target RNA-Directed MicroRNA Degradation. In addition, miRNA can also be stabilized by circRNAs. Hence, selecting miRNA targets based on opposite expression patterns and concluding miRNA sponging by circRNA needs further evidence of direct interactions.

Thank you for your positive comments and kind suggestions.

As suggested by Public Reviewer #1 (12), we employed a biotin-tagged circHMGCS1 to capture miR-4521 and AGO2 in HUVECs (Supplementary Figures 4D and 4E), and Dual luciferase assays have confirmed that miR-4521 can bind to circHMGCS1 directly. Furthermore, RNA pull down and RIP assays have demonstrated the direct binding capability of circHMGCS1 for miR-4521. Collectively, these findings underscore the direct interaction between circHMGCS1 and miR-4521.

(2) The majority of the experiments were performed with an overexpression vector which can generate a lot of linear RNAs along with circRNAs. The linear RNAs produced by the overexpression vectors can have a similar effect to the circRNA due to sequence identity.

In our manuscript, the employed vectors incorporate reverse repeat sequences that facilitate efficient circularization of circRNAs. This design ensures robust circular shearing upon the insertion of circRNA sequences into the polyclonal sites, thereby enhancing the overexpression of circRNAs (Supplementary Figure 2). Moreover, we used lentiviral virus as a vector for circRNA overexpression, not direct plasmid transfection. As demonstrated in Figure 2A, upon overexpression of circHMGCS1, we observed a significant upregulation in circHMGCS1 levels compared to the pLV-circNC and Control groups. Notably, the expression levels of the linear HMGCS1 mRNA did not exhibit significant alterations.

(3) There is a lack of data of circHMGCS1 silencing and its effect on target miRNA & mRNAs.

According to your suggestion, we employed shRNA to knockdown circHMGCS1 in HUVEC, and qRT-PCR was used to assess the expression levels of miR-4521 and ARG1. The knockdown of circHMGCS1 significantly inhibit the expression of circHMGCS1 in HUVEC without obviously affecting the levels of HMGCS1 mRNA. We then selected circHMGCS1 shRNA1 for further investigation. We observed that the knockdown of circHMGCS1 resulted in an upregulation of miR-4521 and a downregulation of ARG1 expression.

**Author response image 1. sa3fig1:** The impact of circHMGCS1 knockdown on ARG1 and miR-4521 expression levels in HUVEC. The cells were transfected with either circHMGCS1 shRNA1 or circHMGCS1 shRNA2, and the expressions levels of circHMGCS1 and HMGCS1 (A), miR-4521 (B) and ARG1 (C and D) in HUVECs were detected by qRT-PCR and Western blot. n=3 in each group. *p < 0.05, **p < 0.01. All significant difference was determined by one-way ANOVA followed by Bonferroni multiple comparison post hoc test, error bar indicates SD.

**Recommendations for the authors:**

**Reviewer #1 (Recommendations For The Authors):**
I suggest improving the discussion based on the literature.(1) Line 131. .... (hsa_circ_0008621, 899 nt in length, identified as circHMGCS1 in subsequent studies because of its host gene being HMGCS1). Please, provide the reference.

We appreciate the valuable comments. We have made changes for improvement, which is add in Line 133(Liang et al, 2021).

(2) The authors conclude that both in vitro and in vivo data suggest that the miR-4521 or circHMGCS1 fails to regulate the effect of diabetes-induced VED in the absence of ARG1. Therefore, ARG1 may serve as a promising VED biomarker, and circHMGCS1 and miR-4521 play a key role in regulating diabetes-induced VED by ARG1. In this context, they should re-evaluate whether this is the best title. "Circular RNA HMGCS1 sponges miR-4521 to aggravate type 2 diabetes-induced vascular endothelial dysfunction"

This manuscript initiates its exploration with circRNA as the focal point of study (Figure 1 and Figure 2), It then delves into the miRNAs associated with circRNA and elucidates their interactions (Figure 3, Figure 4 and Figure 5). Subsequently, the manuscript identifies the target genes of miRNA and validates the regulatory effects of circRNA and miR-4521 on ARG1 (Figure 6). The study culminates with the application of the ceRNA theory to confirm the significance of ARG1 in the functional interplay between circHMGCS1 and miR-4521 (Figure 7). These findings throughout the manuscript are dedicated to uncovering the pivotal roles of circHMGCS1 and miR-4521 in modulating vascular endothelial function. Notably, the interaction between circHMGCS1 and miR-4521 represents a novel discovery of our research. Therefore, we aim to emphasize the critical function of circHMGCS1 and miR-4521 in the regulation of vascular endothelial dysfunction in type 2 diabetes within the manuscript.

**Reviewer #2 (Recommendations For The Authors):**
I have a few suggestions for improving the study further.(1) Although the experiments suggest the role of circHMGCS1, miR-4521 in vascular endothelial function, the direct regulation or interaction of circHMGCS1-miR-4521-ARG1 is unclear. A rescue experiment that checks the effect of circHMGCS1 silencing with/without inhibition of miR-4521 on ARG1 expression must be performed to prove the circHMGCS1- miR-4521 regulatory axis.

Thank you very much for your constructive comments.

According to your suggestion, we utilized shRNA to effectively knockdown circHMGCS1 in HUVEC, Subsequent expression analysis via qRT-PCR was conducted to assess the levels of miR-4521 and ARG1. The knockdown of circHMGCS1 significantly reduced the expression of circHMGCS1 in HUVEC without influencing the expression of the host gene HMGCS1. Concurrently, the knockdown of circHMGCS1 resulted in an upregulation of miR-4521 (Supplementary Figure 4B) and a downregulation of ARG1 (Figure 6P and 6Q). In our manuscript, the upregulation in ARG1 expression caused by circHMGCS1 overexpression was reduced by miR-4521, and the downregulation in ARG1 expression caused by miR-4521 overexpression was also reversed by circHMGCS1. When miR-4521 was knocked down, the expression of ARG1 increased, and circHMGCS1 abrogated its regulatory effect on the expression of ARG1. Collectively, these findings indicate that the interplay between circHMGCS1 and miR-4521 significantly influences ARG1 expression.

**Author response image 2. sa3fig2:** The impact of circHMGCS1 knockdown on ARG1 and miR-4521 expression levels in HUVEC. The cells were transfected with either circHMGCS1 shRNA1 or circHMGCS1 shRNA2, and the expressions levels of circHMGCS1 and HMGCS1 (A), miR-4521 (B) and ARG1 (C and D) in HUVECs were detected by qRT-PCR and Western blot. n=3 in each group. *p < 0.05, **p < 0.01. All significant difference was determined by one-way ANOVA followed by Bonferroni multiple comparison post hoc test, error bar indicates SD.

(2) It is unclear how the authors arrived at the circHMGCS1-miR-4521 pair. The pull down of circHMGCS1 followed by qPCR enrichment analysis of all target miRNAs must be performed to select the target miRNA.

In this manuscript, we identified the expression of miRNA under PAHG treatment through miRNA sequencing, and then further screened out 4 miRNAs with potential binding sites to circHMGCS1 utilizing the miRanda database. Subsequently, we employed qRT-PCR and Western blot analysis to confirm the regulatory influence of miR-4521 on endothelial function (Figure 3). Following this, RIP, RNA pull down, dual luciferase and RNA-FISH experiments were conducted to map the interaction between circHMGCS1 and miR-4521 (Figure 4), the direct interaction between circHMGCS1 and miR-4521 was further substantiated through overexpression and knockdown studies (Figures 5-7). while the reviewer's method may offer a more direct validation, our methodology initially involved a database-driven screening of candidate miRNAs with the potential to target and bind circHMGCS1, followed by experimental validation of these interactions. Both methodologies are capable of establishing the interaction sites between circHMGCS1 and miR-4521.

(3) Since the back splicing is not that efficient, the linear RNA from the overexpression construct may produce many linear RNAs with miRNA binding sites. The effect seen in the case of overexpression experiments needs to consider the level of linear and circular HMGCS1 produced by the vector.

In this manuscript, the vector's multiple cloning site is flanked by inverted repeat sequences that facilitate efficient circRNA looping. This design enables the inserted sequence to form a stable loop and undergo circularization upon transcription, leading to the overexpression of circRNA (Supplementary Figure 2). For the validation of circular RNA, we employed divergent primers that straddle the circRNA splicing junction. These primers are specific for circRNA amplification and do not amplify the corresponding linear RNA, as demonstrated in Figure 2A. Upon overexpression of circHMGCS1, we observed a significant increase in circHMGCS1 levels compared to the empty vector and Control groups, while there was no significant change in the expression level of HMGCS1 mRNA.

(4) As miR-4521 has multiple miRNA binding sites on circHMGCS1, it is not very clear which sites were mutated in circHMGCS1-MUT.

We have made corrections to Supplementary Figure 4C. Utilizing the miRanda algorithm, we identified 10 potential binding sites for miR-4521 on circHMGCS1. Subsequently, we selected the site with the highest binding affinity for mutational analysis (miR-4521 binding positions 3-15, circHMGCS1 binding positions 260-281, binding rate 91.67%, binding ability -17.299999 kCal/Mol). We employed a dual-luciferase assay to confirm the direct interaction between circHMGCS1 and miR-4521.

(5) Since the ceRNA network works efficiently in an equimolar concentration of the regulatory molecules, providing the copy number of circHMGCS1, miR-4521, and target mRNAs would be helpful.

We employed qRT-PCR to ascertain the absolute quantification of mRNA copy numbers, following established methodologies (Nolan et al, 2006; Wagatsuma et al, 2005; Zhang et al, 2009). Our qRT-PCR data reveal that the circHMGCS1 mRNA copy number is 2343±529. In comparison, the ARG1 mRNA copy number stands at 88±27, while the miR-4521 copy number is significantly higher, recorded at 36277±9407.

**Author response image 3. sa3fig3:** The distribution of copy numbers for circHMGCS1, miR-4521 and ARG1 in HUVECs.

(6) The yellow highlighted "cyclization-mediated sequence-F & R" does not seem to be complementary sequences. The method section may include the details of the vectors and cloning strategies for the overexpression constructs.

The figure below illustrates the schematic representation of the complementary structure between the upstream and downstream sequences that facilitate circRNA circularization. This strategic pairing is designed to enhance the circularization efficiency of circRNA while concurrently suppressing mRNA synthesis (Liang & Wilusz, 2014). Details of this design have been integrated into the experimental method (Line539). The specific additions are as follows:

The circHMGCS1 sequence [NM_001098272: 43292575-43297268], the splice site AG/GT and ALU elements were inserted into the pCDH-circRNA-GFP vector (upstream ALU: AAAGTGCTGAGATTACAGGCGTGAGCCACCACCCCCGGCCCACTTTTTGTAAAGGTACGTACTAATGACTTTTTTTTTATACTTCAG, downstream ALU: GTAAGAAGCAAGGAAAAGAATTAGGCTCGGCACGGTAGCTCACACCTGTAATCCCAGCA). The restriction enzyme sites selected were EcoRI and NotI.

**Author response image 4. sa3fig4:** 

(7) Since circHMGCS1 is a multi-exonic circRNA that can undergo alternative splicing and divergent primers only validate the backsplice junction, the full-length sequence of mature circHMGCS1 needs to be checked by circRNA-RCA PCR followed by Sanger sequencing.

In compliance with your guidance, we have enriched the revised manuscript with additional data. Specifically, we have included the full-length nucleic acid electrophoresis diagram of circHMGCS1 in Supplementary Figure 1F, the Sanger sequencing results in Supplementary Figure 1G, and a comparative analysis of the circHMGCS1 sequences obtained from Sanger sequencing with those referenced in the circBase database, presented in Supplementary Figure 1H.

Reference:

Barbon, E., C. Kawecki, S. Marmier, A. Sakkal, F. Collaud, S. Charles, G. Ronzitti, C. Casari, O.D. Christophe, C.V. Denis, P.J. Lenting, and F. Mingozzi. 2023. Development of a dual hybrid AAV vector for endothelial-targeted expression of von Willebrand factor. Gene Ther. 30: 245-254.

Fu, Y., J. Chen, and Z. Huang. 2019. Recent progress in microRNA-based delivery systems for the treatment of human disease. ExRNA. 1: 24.

Kristensen, L.S., M.S. Andersen, L.V.W. Stagsted, K.K. Ebbesen, T.B. Hansen, and J. Kjems. 2019. The biogenesis, biology and characterization of circular RNAs. Nat Rev Genet. 20: 675-691.

Liang, D., and J.E. Wilusz. 2014. Short intronic repeat sequences facilitate circular RNA production. Genes Dev. 28: 2233-2247.

Liang, J., X. Li, J. Xu, G.M. Cai, J.X. Cao, and B. Zhang. 2021. hsa_circ_0072389, hsa_circ_0072386, hsa_circ_0008621, hsa_circ_0072387, and hsa_circ_0072391 aggravate glioma via miR-338-5p/IKBIP. Aging (Albany NY). 13: 25213-25240.

Liu, C.X., and L.L. Chen. 2022. Circular RNAs: Characterization, cellular roles, and applications. Cell. 185: 2016-2034.

Nolan, T., R.E. Hands, and S.A. Bustin. 2006. Quantification of mRNA using real-time RT-PCR. Nat Protoc. 1: 1559-1582.

Wagatsuma, A., H. Sadamoto, T. Kitahashi, K. Lukowiak, A. Urano, and E. Ito. 2005. Determination of the exact copy numbers of particular mRNAs in a single cell by quantitative real-time RT-PCR. J Exp Biol. 208: 2389-2398.

Yao, C., T. Veleva, L. Scott, Jr., S. Cao, L. Li, G. Chen, P. Jeyabal, X. Pan, K.M. Alsina, I.D. Abu-Taha, S. Ghezelbash, C.L. Reynolds, Y.H. Shen, S.A. Lemaire, W. Schmitz, F.U. Müller, A. El-Armouche, N. Tony Eissa, C. Beeton, S. Nattel, X.H.T. Wehrens, D. Dobrev, and N. Li. 2018. Enhanced Cardiomyocyte NLRP3 Inflammasome Signaling Promotes Atrial Fibrillation. Circulation. 138: 2227-2242.

Zhang, X.X., T. Zhang, M. Zhang, H.H. Fang, and S.P. Cheng. 2009. Characterization and quantification of class 1 integrons and associated gene cassettes in sewage treatment plants. Appl Microbiol Biotechnol. 82: 1169-1177.

Zincarelli, C., S. Soltys, G. Rengo, and J.E. Rabinowitz. 2008. Analysis of AAV serotypes 1-9 mediated gene expression and tropism in mice after systemic injection. Mol Ther. 16: 1073-1080.